# Outcomes of a 12-week ecologically valid observational study of first treatment with methylphenidate in a representative clinical sample of drug naïve children with ADHD

Kristine Kaalund-Brok[1‡¶]*, Tine Bodil Houmann[1‡¶], Marie Bang Hebsgaard[1☯], Maj-Britt Glenn Lauritsen[1☯], Louise Hyldborg Lundstrøm[1☯], Helene Grønning[1☯], Lise Darling[1☯], Susanna Reinert-Petersen[1☯], Morten Aagaard Petersen[2‡], Jens Richardt Møllegaard Jepsen[1,3‡], Anne Katrine Pagsberg[1‡¶], Kerstin Jessica Plessen[1‡¶], Henrik Berg Rasmussen[4,5‡¶], Pia Jeppesen[1‡¶], INDICES[¶]

1 Child and Adolescent Mental Health Centre, Research Unit, Mental Health Services, The Capital Region of Denmark, Hellerup, Denmark, 2 The Research Unit, Department of Palliative Medicine, Bispebjerg Hospital, Copenhagen University Hospital, Copenhagen, The Capital Region of Denmark, Copenhagen, Denmark, 3 Center for Clinical Intervention and Neuropsychiatric Schizophrenia Research (CINS) and Center for Neuropsychiatric Schizophrenia Research (CNSR), Mental Health Services Glostrup, The Capital Region of Denmark, Glostrup, Denmark, 4 Institute of Biological Psychiatry, Mental Health Centre Sct. Hans, Copenhagen University Hospital, The Capital Region of Denmark, Roskilde, Denmark, 5 Department of Science & Environment, Roskilde University, Roskilde, Denmark

☯ These authors contributed equally to this work.
‡ KK-B, TBH, MAP, JRMJ, AKP, KJP, HBR and PJ also contributed equally to this work.
¶ Membership of the INDICES Consortium is listed in the Acknowledgments.
* kristine.kaalund-brok@regionh.dk

**Data Availability Statement:** The study measured the outcomes of a 12-week ecologically valid observational study of first treatment with

## Abstract

Randomized placebo-controlled trials have reported efficacy of methylphenidate (MPH) for Attention-deficit/hyperactivity disorder (ADHD); however, selection biases due to strict entry criteria may limit the generalizability of the findings. Few ecologically valid studies have investigated effectiveness of MPH in representative clinical populations of children. This independently funded study aims to describe treatment responses and their predictors during the first 12 weeks of MPH treatment using repeated measurements of symptoms and adverse reactions (ARs) to treatment in 207 children recently diagnosed with ADHD. The children were consecutively included from the Child and Adolescent Mental Health Centre, Mental Health Services, The Capital Region of Denmark. The children (mean age, 9.6 years [range 7–12], 75.4% males) were titrated with MPH, based on weekly assessments of symptoms (18-item ADHD-rating scale scores, ADHD-RS-C) and ARs. At study-end 187 (90.8%) children reached a mean end-dose of 1.0 mg/kg/day. A normalisation/borderline normalisation on ADHD-RS-C was achieved for 168 (81.2%) children on the Inattention and/or the Hyperactivity-Impulsivity subscale in week 12, and 31 (15.0%) children were non-responders, which was defined as absence of normalisation/borderline normalisation ($n$ = 19) or discontinuation due to ARs ($n$ = 12), and eight (3.8%) children dropped out from follow-up. Nonresponders were characterised by more severe symptoms of Hyperactivity-Impulsivity and global impairment before the treatment. ARs were few; the most prominent

methylphenidate in a representative clinical sample of drug naïve children with ADHD. The study was a part of the routine care in the clinic. The data were both documented in the medical record at the Child and Adolescent Mental Health Centre and in an anonymized data file with a separate key file. All data were sensitive patient information. The pseudonymous individual participant data that underlie the results reported in this article, (text, tables, figures, and appendices) can be made available to investigators for individual participant data meta-analyses that have been approved by independent review committees. The data access will be granted on a case-by-case basis by the principal investigator (Tine Bodil Houmann) and the non author point of contact, the data manager Michella Heinrichsen after further approval by the Capital Region of Denmark, Copenhagen, Denmark. Access will be granted to the extent permissible by the General Data Protection Regulation and the Danish Data Protection Act. Making the data available may require approval from the Danish Data Protection Authority. The pseudonymous data can be made available from six months after the publication date of this Article, and with no end date. Proposals for use of data and requests for access should be directed to tine.bodil.houmann@regionh.dk or michella.heinrichsen@regionh.dk. To gain access, researchers will need to sign a data access agreement with the Research Unit of the Child and Adolescent Mental Health Centre - Capital Region of Denmark, Copenhagen, Denmark. The study was registered in ClinicalTrials.gov (NCT04366609). The study was approved by the Danish Data Protection Agency (P-2019-851). The Local Committee on Health Research Ethics was consulted (J.nr. H-B-2009-026), and the study was evaluated not be within their jurisdiction due study design as an observational study.

**Funding:** The Danish Council for Strategic Research, Programme Commission on Individuals, Disease and Society (grant 10–092792/DSF) funded the INDICES study. Website: https://ufm.dk/en/research-and-innovation/councils-and-commissions/former-councils-and-commissions/the-danish-council-for-strategic-research?set_language=en&cl=en. The funders had no role in study design, data collection and analysis, decision to publish, or preparation of the manuscript. KKB and HBR has been paid by Danish Council for Strategic Research for their work in INDICES. TBH and PJ has been paid very little for participating in INDICES. The rest of the authors has not been paid for participating. The funders had no role in study design, data collection and analysis, decision to publish, or preparation of the manuscript.

were appetite reduction and weight loss. A decrease in AR-like symptoms during the treatment period questions the validity of currently available standard instruments designed to measure ARs of MPH. This ecologically valid observational study supports prior randomized placebo-controlled trials; 81.2% of the children responded favourably in multiple domains with few harmful effects to carefully titrated MPH.

**Clinical trial registration:** ClinicalTrials.gov with registration number NCT04366609.

## Introduction

Attention deficit/hyperactivity disorder (ADHD) as defined in the DSM-IV/-5 (Diagnostic and Statistical Manual of Mental Disorders, DSM) [1] is a heterogeneous neurodevelopmental disorder characterised by pervasive and impairing symptoms of inattention and/or hyperactivity and impulsivity with onset of symptoms before age 7 (DSM-IV) or age 12 (DSM-5). The syndrome of Hyperkinetic disorder based on the International Classification of Diseases and Related Health Problems (ICD-10) [2] is roughly equivalent to the combined presentation of ADHD as per the DSM-IV/-5, and regardless of definitions, the ADHD syndromes show high rates of persistence into adulthood [3, 4]. The mean estimated worldwide prevalence of ADHD is 3.4% (CI 95% 2.6 to 4.5) in the general population of children and adolescent, making it the most common neurodevelopmental disorder in youth [5]. ADHD has a polygenic and multifactorial aetiology that is only partly understood [6].

The National Institute for Health and Care Excellence (NICE; United Kingdom) recommends methylphenidate (MPH) as a first-line pharmacological treatment of ADHD in children [7], and recent guideline strongly recommend parent management training and other behavioural treatments within family and school along with medication [8, 9]. MPH is a central nervous system stimulant that has been used for treatment of ADHD in youth since the 1960s [10]. Although its mechanism of action is not completely understood, MPH inhibits the dopamine and the norepinephrine transporters [11], primarily resulting in increased extracellular levels of dopamine in the brain [12, 13].

Several meta-analyses of the short-term efficacy of immediate release MPH (IR-MPH) in randomized placebo-controlled trials have reported large effect sizes (range from 0.54 to 1.78) on ADHD core symptoms, when effects are measured as differences in endpoint or change in scores of parent-rated and teacher-rated ADHD symptoms and behaviours [14, 15]. A review of treatments for ADHD in adolescents aged 12–18 years found large effects for stimulants (extended-release MPH and amphetamine) describing a mean change in absolute symptoms score ranging from 10 to 18 points on the clinician-rated, 18-item-ADHD-rating-scale (ADHD-RS-C, range 0–54 [16, 17]). Also, two network metaanalyses have reported favorable efficacy of MPH compared with placebo in randomized placebo-controlled trials [18, 19].

Beneficial effects of IR-MPH have been demonstrated in several domains including improvements in ADHD symptoms and behaviours at home and in school [14, 20], cognitive functions [21], classroom behaviour, and academic performance [22], although a recent meta-analysis found only small effects on school performance [23]. The primary measure of MPH efficacy in randomized controlled trials (RCTs) is usually the endpoint or change in scores of total ADHD symptoms, and clinically meaningful response is generally considered to be a within-group symptom reduction of 25% [24] or 30% [25] on ADHD-RS (corresponding to 10–15 absolute points).

**Competing interests:** The authors have declared
that no competing interests exist.

However, there is no generally accepted definition of the clinically significant response to
treatment for ADHD, and the response rates obviously vary with the psychometric instrument,
the informant, and the defined response criteria [15, 24]. In one RCT, the response was
defined as a 40% or more decrease in sum scores in ADHD-RS-C after 6 weeks of treatment
[26], resulting in a response rate of 64% in youth patients treated with MPH for the first time.

Generally, adverse reactions (ARs) to MPH treatment are poorly described in the literature,
partly due to huge variations in the definitions and the instruments for measurement of ARs
[27]. A review of ARs associated with MPH in short-term studies showed rates of loss of appe-
tite of 3 to 56%, poor sleep of 9 to 64%, headache of 2 to 33%, and abdominal pain of 4 to 19%
[27]. Another review of ARs induced by MPH in long-term studies with a duration of one to
two years reported discontinuation rates due to ARs of 8 to 15%, and incidences of any ARs of
85 to 89%, loss of appetite 14 to 19%, poor sleep 15 to 19%, headache 25 to 30%, and abdominal
pain 8 to 11% [28]. Although the efficacy and safety of MPH is well documented in placebo-
controlled RCTs and meta-analyses of RCTs, less is known about the "real world" effectiveness
of MPH for treatment of heterogeneous samples of children with ADHD in service of child
and adolescent mental health. The RCTs of MPH may be biased by selection according to the
studies' inclusion and exclusion criteria (typically excluding comorbidity), the patients' will-
ingness to participate, and the clinicians' willingness to allocate patients to randomised inter-
vention studies. The limited generalizability of the evidence from the RCTs, including the risk
of selection bias, calls for more naturalistic observational studies of the effectiveness of MPH in
representative clinical samples of children with ADHD, treated in routine settings, including
all MPH-treated children regardless of ADHD severity, ADHD subtype and comorbidity [29].

In preparation for the analysis of the present study, we systematically searched PubMed for
naturalistic observational clinical studies of MPH treatment of children with ADHD including
studies from the start of the PubMed database from 1987 to January 6, 2020 (S1 Fig, PRISMA
flow chart; S1 Table, criteria for literature search). A total of 43 publications covering 30 natu-
ralistic observational MPH studies were identified of MPH-free (no MPH treatment at the time
of inclusion) children aged 7–12 years and of these were 21 publications covering 17 studies of
MPH naïve children aged 6–18 years, and of these were 11 studies with 14 publications with
subgroup analyses (e.g. outcomes of specific genotypes) or specific neurocognitive outcome
[30–43] (S2 Table, included studies). The 11 studies of MPH naïve children included a total of
$n$ = 1537 patients (51–280 per study) with a mean age of 9.3 [mean range 8.0–10.3] years. The
mean end-dose of MPH was 0.80 mg/kg/day [range 0.50–1.06]. The mean follow-up time was
9.5 weeks [range 4–24], with a number of follow-up points in time varying from 1 to 6 weeks.

Only nine studies monitored the response to MPH using well-known psychometric instru-
ments: DuPaul's ADHD Rating Scale rated by parents (ADHD-RS-P, [range 0–54]) [16] and
rated by clinicians (ADHD-RS-C [range 0–54]) [17], the revised Conner's Parent Rating Scale
(CPRS-R, [range 0–144]) [44], Swanson Nolan Pelham version IV scale (SNAP-IV, [range
0–60] [45], Test of Variables of Attention (TOVA) [46], Clinical Global Impression Improve-
ment (CGI-I, [range 1–7]), and/or CGI Severity (CGI-S, [range 1–7]) [47]. Nine studies (11
publications) a priori defined a response criterion for symptom reduction measured with vali-
dated instruments and the reported response rates varied from 15.1% to 81.5% [30–33, 35, 37,
41–43]. The mean reduction of ADHD core symptoms (in percentage of symptom level at
entry) varied from 23.8% to 62.7% [31–33, 35, 39, 40]. Only one study divided ADHD-RS-P
and ADHD-RS-C into subscales of Inattention [range 0–27] and Hyperactivity-Impulsivity
[range 0–27], and the mean reductions were 41.4% to 59.3% on the Inattention and 47.3% to
66.0% on the Hyperactivity-Impulsivity subscales, respectively [33]. Seven of the 11 studies
apparently monitored ARs [31, 33, 34, 36, 39, 48], but the reporting was incomplete and thus
difficult to summarise. Reduced appetite was present in roughly 25–66% of patients [33, 34,

39, 42] using Barkley's Stimulant Side Effect Rating Scale [49] (BSSERS) and severe appetite reduction in 21.3% of patients [48]. Weight loss was reported in 34% of patients [39]. Intolerable ARs (defined by each study) were found in 8% of patients [31]. The vast majority of patients (75–82%) had one or more ARs during the treatment period [33, 34]. One study measured 82.8% of patients having more than one AR during the treatment period while only 4.5% of patients withdrew from the study due to ARs [33].

In summary, there is no consensus on the definition of a clinically significant response to MPH treatment, and the reported response rates, and rates of ARs apparently vary with patient sample, study design, and the applied psychometric instruments and informants.

The overall objective of this ecologically valid prospective observational intervention study was to characterise the beneficial effects and ARs of IR-MPH treatment in a clinical sample of MPH naïve children recently diagnosed with ADHD and offered an individually titrated dosing of IR-MPH. The specific aims were to (a) describe the treatment response during the first 12 weeks after initiation of IR-MPH treatment based on weekly clinician-rated ADHD core symptoms and behaviours, the rate of normalisation/borderline normalisation of ADHD core symptoms, ARs, daily and social functioning, and indices of sustained attention; (b) provide information about the predictive value of clinical characteristics at entry (sex, age group, global severity of psychiatric disorder, psychiatric comorbidity, and subtype of ADHD diagnoses) for these outcomes; and (c) determine the end-dose of IR-MPH.

## Materials and methods

### Participants

The study included MPH naïve boys and girls aged 7–12 years with a recent ICD-10 diagnosis of hyperkinetic disorder (F90.0–90.9) or attention deficit disorder without hyperactivity (F98.8) and clinical indication for treatment with IR-MPH. Patients referred to the Child and Adolescent Mental Health Centre, Mental Health Services (The Capital Region of Denmark) in the period from 1st of May 2012 to 1st August 2014, and who were suspected of having ADHD were consecutively screened for study eligibility.

The exclusion criteria were mental retardation (ICD-F70.X or IQ < 70), previous treatment with drugs metabolised by carboxylesterase 1 (CES1) (see below); e.g., MPH, severe comorbid psychiatric or somatic disease that resulted in contraindication for treatment with MPH (e.g., schizophrenia or cardiac disease), language barriers, and lack of informed consent.

### Study design

The study was a prospective, noncontrolled intervention study. It is a part of the Danish INDI-CES study as work package six (INDIvidualised drug therapy based on pharmacogenetics: focus on personalising the treatment of drugs metabolised by carboxylesterase 1 (CES1) [50]. Study protocol is available (S1 and S2 Files). The study followed the protocol, and the only deviations from the protocol was that we did not include CGI-Efficacy [47] and ASK-ME attitude [51].

### Diagnostic evaluation

The diagnostic procedure was performed in accordance with the Danish "National clinical guideline for the assessment of ADHD in children and adolescents" [52]. A diagnosis of ADHD was posed after a thorough assessment including clinical observations and examinations of the children using the following instruments: Schedule for Affective Disorders and Schizophrenia for School-Age Children, Present and Lifetime Version [53], ADHD-RS-P and

ADHD-RS-T [16, 54, 55], parent rated Child Behaviour Checklist and teacher rated Teacher Report Form [56–58], Weiss Functional Impairment Rating Scale–Parent version (WFIRS-P) [59, 60], and TOVA [46, 61]. Experienced medical doctors and psychologists, all trained in the instruments used in this study, conducted all assessments.

As a part of the routine clinical care, a senior consultant in child and adolescent psychiatry confirmed the ADHD diagnosis, any psychiatric comorbid disorders, and the indication for treatment with IR-MPH. Patients were consecutively included in the study after their parents had given informed consent to participate in the study.

## Administration of IR-MPH treatment

According to the Danish guidelines for treatment of ADHD, MPH is the first-line drug when pharmacological treatment is indicated. The Danish guidelines are similar to The NICE guidelines: use of an initial low oral dose of MPH and an up-titration period of at least 4 weeks, until no further effect is measured on a standard ADHD rating scale, or the appearance of intolerable ARs, or a maximum dose of 2.1 mg/kg/day [7, 62]. Based on these guidelines, we developed an enhanced IR-MPH-treatment manual for the present study with weekly telephone-based ratings and monthly clinical assessments of symptom change and the presence of ARs to allow for an individually titrated optimal dosing of IR-MPH. The study used IR-MPH in the form of Medikinet® 5, 10 and 20 mg because of the option to split the tablet. It was the children's regular clinicians who decided to treat each patient with IR-MPH as a part of the routine care. The initial dose was 2.5 or 5 mg IR-MPH depending upon a patient's weight ($</>$ 30 kg). Tablets were given two or three times a day as determined by each patient's needs and was administered of parents and teachers. The weekly individually dose increment of 2.5 or 5 mg IR-MPH per dose continued until a clinically significant response, defined as normalisation/borderline normalisation on ADHD-RS-C was achieved, ARs prohibited further up-titrations of the total daily dose of IR-MPH, or the maximum dose was reached. Some circumstances allowed patients to change to other MPH formulations (Motiron® in the event of lactose intolerance or extended-release-MPH if patients were non-compliant or had ARs due to IR-MPH). Patients were excluded from further study assessments if their medication changed to a non-MPH preparation (e.g., atomoxetine) or if they were noncompliant with the treatment.

## The assessment programme

The patients were enrolled in the study for a 12-week study period from 1st of May 2012 to 1st August 2014. After enrolment, the initial assessments were done immediately before initiation of IR-MPH in week 0. The clinical investigator (KKB, TH, and MH) undertook the weekly evaluation of ADHD core symptoms and ARs using the ADHD-RS-C and the clinician-rated BSSERS-C. The weekly evaluations were based on telephone interviews with the parent in week 1, 2, 3, 5, 6, 7, 9, 10, 11, and on more thorough clinical evaluation at baseline (week 0) and in week 4, 8, and 12, including interviews with the parents about the children's daily functions and symptoms, clinical observation and physical examination of the patients, and parent- and teacher-rated ADHD core symptoms (ADHD-RS-P and ADHD-RS-T). The clinical investigator and the children's regular clinicians who all used the ADHD-RS-C in week 0 and 12 conducted consensus ratings of the children's ADHD core symptoms. Decisions about changes in IR-MPH dosing or discontinuation due to ARs were made in collaboration with the children's' regular clinicians and the families at the study site. The clinical investigator and the children's regular clinicians where not blind to the dosage of IR-MPH, as they collaborated to personalise the dosing of MPH based on the weekly assessments of the ADHD core symptoms and ARs for the treatment of each child.

The same pair of the clinical investigator and the child's regular clinician followed the same patient over 12 weeks (the first author was the clinical investigator and rater in 71% of cases). In weeks 0 and 12, each rater performed an independent rating of the ADHD-RS-C before they agreed on a consensus rating. The two independent ratings were used for calculation of the interrater reliability for the pair at each time point using the intraclass correlation coefficient (ICC). The mean of the ICCs was calculated for each pair across time points. The mean ICCs ranged from 0.72 to 0.99 (mean of ICC = 0.90). If any disagreement was found, the single items of ADHD-RS-C were discussed until consensus was reached. Consensus ratings of ADHD-RS-C scores in week 0 and 12 were measured to set an individual standard for the clinical investigator ratings in week 1 to 11. None of the clinicians were blinded to evaluation of the treatment and changes of the treatment with MPH.

## Assessment instruments

**ADHD-RS-scales.** The primary outcome of the 12-week IR-MPH-treatment was measured with the ADHD-RS scales [16, 17, 54, 55]. The ADHD-RS-P and the ADHD-RS-T each consist of nine questions of inattention, nine questions of hyperactivity and impulsivity, and six questions of conduct problems evaluated on a four-point Likert scale from 0 (*none = never or rarely*) to 3 (*severe = very often*). The 18-item, clinician-rated ADHD-RS (ADHD-RS-C) is identical to the ADHD-RS-P and ADHD-RS-T subscales measuring Inattention and Hyperactivity-Impulsivity, whereas Conduct problems are not scored [17]. ADHD-RS-P and ADHD-RS-T have previously been validated in a Danish general population-based sample (*n* = 865 children) [55], and the standardised scores (*t-scores*) stratified for each sex and age group (7–9 and 10–12 years) were used to delineate the cut-off for normalisation ($\leq$ 60 *t-scores*) or borderline normalisation (60–70 *t-scores*) on the clinician-rated ADHD-RS-C scores of Inattention and of Hyperactivity-Impulsivity in the present study. High correlations between ADHD-RS-C and ADHD-RS-P have been documented [63]. In the present study, ADHD-RS-C scores were determined using all available information including interviews with the parents, the parent- and teacher-rated ADHD-RS scores, psychiatric assessment, and observation of the children. The ADHD-RS-C scores were used to describe change in symptoms (sum scores) and proportions of children normalised (Nor) or borderline normalised (Bnor); any treatment response was defined as Nor or Bnor (Nor/Bnor).

**Clinical Global Impression Severity scale (CGI-S).** The clinician-rated CGI-S (English version) is a seven-point Likert scale, rated 1 (*normal, not ill at all*) to 7 (am*ong the most extremely ill patients*), used to measure the global severity of symptoms and functional impairments of mental health disorder including ADHD and psychiatric comorbidities [47]. Consensus ratings were performed in week 0 and 12. A beneficial symptom reduction was defined by a CGI-S score of 1 or 2 (*normal, not ill at all* or *borderline ill*) based on a definition from another study [32].

**Clinical Global Impression Improvement (CGI-I).** The clinician-rated CGI-I (English version) is also a seven-point Likert scale rated from 1 (*very much improved*) to 7 (*very much worse*) [47]. Consensus ratings were performed in week 12. A beneficial symptom reduction was defined by a CGI-I score of 1 or 2 (*very much improved* or *much improved*) based on a definition from other studies [35, 41, 42].

## Test of Variables of Attention (TOVA)

TOVA is a computerized, continuous performance test and is validated to evaluate the objective response to the treatment of ADHD [46, 61, 64, 65]. TOVA was administrated in week 0 and 12, and number of commission errors (response to non-target), number of omission

errors (non-response to target), the response time in microseconds, and the variability of response time in correct responses to target were measured over the 21.6-minute-long test used as total raw scores. The study used version 7.3 and 8.0 of TOVA [66, 67]. Consistently, IR-MPH was dosed at 1 p.m. and TOVA was performed at 2 p.m. [68] to ensure an equal and sufficient exposure of IR-MPH at the week 12-assessment (maximal plasma concentration is achieved 1–2 hours after administration of IR-MPH [69]). In week 0 and 12, total raw scores were converted to *z*-scores based on mean and standard deviation of week 0 raw scores due to the lack of standard of TOVA for Danish children.

**Weiss Functional Impairment Rating Scale—Parent version (WFIRS-P).** WFIRS-P (Danish version) is a parent-rated questionnaire, with 50 questions about a child's daily and social functioning in six different domains of a child's life: family (10 items), learning and school (10 items), activities of daily living (10 items), self-concept (3 items), social activities (7 items), and risky activities (10 items) [59, 60, 70]. It is evaluated on a four-point Likert scale from 0 (*never or not at all*) to 3 (*very often or very much*). Index scores of WFIRS-P were measured in week 0 and 12 [71]. Single item scores = 2–3 signal impairment (> 2 SD) [59, 60, 70]. If the parent-reported scores were missing for a single item on a subscale (maximum 1 item) we still calculated the index score as the sum-score divided by the number of items that were scored.

**Barkley's Stimulant Side Effect Rating Scale (BSSERS-C).** ARs were measured by the clinical investigator using the BSSERS-C (Danish version) [49]. It consists of 17 effect items rated by the clinician, based on information from patients, parents, and clinical observations: insomnia, nightmares, staring, talks less, disinterested in others, reduced appetite, irritable, stomachaches, headaches, drowsiness, sadness, prone to crying, anxious, nail biting, euphoria, dizziness, and tics/nervous movements. BSSERS-C is a 10-point Likert scale rated from 0 (*problem absent*) to 9 (*problem evokes serious impairment*). A manual for interviewing and rating using BSSERS-C was elaborated for the study, anchoring the 10-point problem scores (e.g., specifying the time to fall asleep when scoring insomnia). The BSSERS-C was scored before initiation of IR-MPH-treatment (week 0) and weekly during treatment. Severities of ARs were calculated as the sum of problem scores on the 17-item BSSERS-C. Significant changes in single items were also explored (e.g., reduced appetite; 1 item, range 0–9).

**Dosing of IR-MPH.** The IR-MPH doses were individually titrated, based on the weekly evaluations of symptom reductions and ARs (and the body weight of a child), which were carried out by the clinical investigator. The dose of IR-MPH (mg/kg/day) at the end of the study was registered.

**Physical assessments.** The body weight, height, heart rate and blood pressure were measured every fourth week with the same equipment each time. The study followed the European guidelines [72] and measured blood pressure after 10 minutes of rest with a sphygmomanometer (nonelectrical) three times and the average of the last two measures was used.

**Nonresponder.** A combined measure of nonresponder status was defined as patients who discontinued IR-MPH treatment due to ARs or serious adverse reactions (SARs) or patients who did not attain Nor/Bnor status, based on the clinical experience that these two types of poor outcome are interrelated.

## Predictors

We explored whether the following clinical baseline characteristics were predictors for outcomes:

Age (7–9 or 10–12 years), sex, comorbidity (two/more or none/one comorbid psychiatric diagnosis), intelligence level (IQ 70–85 (*inferioritas intellectualis*) or IQ > 85), conduct disorder (ICD10 DF90.1/DF91.X/DF92.X or no conduct disorder), CGI-S in week 0 as a continuous

variable [range 1–7] or dichotomized to *normal not ill* to *markedly severity* of psychiatric disease (CGI-S 1–5) and *severe* to *extreme among the most ill patients* (CGI-S 6–7), ADHD diagnosis (ICD10 DF90.0/ DF90.8/DF90.9 or DF90.1 or DF98.8), and weight in week 0 ($\leq$ 30kg or > 30kg).

The primary outcomes measures were: (a) the number of patients (percentage) who obtained Nor (*t-score* $\leq$ 60) or Bnor ($\leq$ 60 *t-score* $\leq$ 70) of the ADHD-RS-C scores of Inattention and of Hyperactivity-Impulsivity at week 12; (b) the course of the weekly ADHD-RS-C scores of Inattention and of Hyperactivity-Impulsivity during week 0–12; (c) the course of the weekly BSSERS-C scores of ARs during week 0–12; and (d) the end-dose IR-MPH.

## Statistical analyses

The study was powered to detect effects of genetic variants on outcome regarding the end-dose of IR-MPH = 1 mg/kg/day (SD 0.5 mg/kg/day) and the average MPH titration duration = 6 weeks (SD 2 weeks). Setting a significance level of 0.05 (one-sided for genetic variants) and a power of 0.80, then a minimum relevant effect measured as a regression-coefficient of $\beta$ = 0.25 (mg/kg/day) or $\beta$ = 0.1 (weeks) would be detectable in 94 children with a frequency of 0.20 of the CES1 variant of interest. It was decided to include at least 200 children to be able to study less frequent variants with a relatively large impact on the outcome [73].

Comparisons of baseline characteristics of included and excluded patients were carried out with the independent *t*-test for continuous variables and Chi square test for categorical variables.

Changes from week 0 to week 12 in scores on ADHD-RS-C subscales, ADHD-RS-P subscales, ADHD-RS-T subscales, WFIRS-P subscales, the BSSERS-C sum score, height, weight and blood pressure were analysed with paired *t*-test for continuous variables and $X^2$ test for categorical variables. Missing data on any items were not allowed on ADHD-RS-C, ADHD-RS-P, ADHD-RS-T and BSSERS-C. Ten percent missing data on items on WFIRS-P subscales were allowed and missing data on items were set as 0 (*never or not at all*). Changes in *z*-scores and raw scores of response time, variability of response time, omission errors, and commission errors of TOVA were tested with paired *t*-test.

To explore the predictive value of clinical baseline characteristics on the responses of 12 weeks of treatment with IR-MPH, linear mixed effect models for repeated measures were used. Unstructured covariance type fitted the data best as judged by the Akaike information criterion (compound symmetry covariance was used in case a model with unstructured covariance was impossible to fit). The outcome variables of the mixed models were the sum scores of Inattention and Hyperactivity-Impulsivity (ADHD-RS-C) and the sum score of ARs (BSSERS-C). The mixed models of Inattention and Hyperactivity-Impulsivity (ADHD-RS-C) tested the potential impact of explanatory variables of age, sex, intelligence level, conduct disorder, comorbidity, and time. A mixed model of ARs (BSSERS-C) tested the impact of age, sex, intelligence level, conduct disorder, comorbidity, time, ADHD diagnosis, and CGI-S in week 0. Time was measured as a continuous variable coded 0, 1, 2...12 weeks, and all other explanatory variables were measured as categorical variables. It was problematic to estimate the full mixed models including all explanatory variables simultaneously. Therefore, the explanatory variables were first tested pairwise (as main factors and interactions) and only significant main factors and interactions were included in the final multiple mixed model analyses.

Repeated measures of reduced appetite (single item of BSSERS-C) were not normally distributed. Reduced appetite was dichotomised into mildly reduced or no reduced appetite (BSSERS-C single score $\leq$ 3) and moderately to severely reduced appetite (BSSERS-C single score $\geq$ 4). To identify potential predictors of reduced appetite over 12 weeks, generalized estimation equations (GEEs) were used. The correlation structure for the model was unstructured

or M-independent. It was problematic to estimate the full GEEs including all explanatory variables simultaneously. Therefore, as described above, variables were first tested pairwise, and only significant main factors and interactions were included in the final multiple GEE analyses. GEEs of reduced appetite tested the impact of age, sex, intelligence level, conduct disorder, comorbidity, time, ADHD diagnosis, and CGI-S in week 0. The ADHD disorders were categorised into ICD-10 hyperkinetic disorder or hyperkinetic conduct disorder (DF90.0, DF90.8, DF90.9, and DF90.1) versus attention deficit disorder without hyperactivity (DF98.8). CGI-S in week 0 was dichotomised to *not ill* to *markedly severity* of psychiatric disease (CGI-S 1–5) and *severe* to *extreme among the most ill patients* (CGI-S 6–7).

To determine independent predictors for the chance of normalisation or borderline normalisation on ADHD-RS-C Inattention and Hyperactivity-Impulsivity, Cox regression analysis with backward elimination was used. The explanatory variables in the Cox regression model were age, sex, cognitive deficits, conduct disorder, and comorbidity (categorical variables). Dropouts were censored out when patients exited the study.

The association between the end-dose of IR-MPH per day at week 12 and the severity of the psychiatric disorder (CGI-S week 0) was explored using linear regression with backward eliminations, with adjustment for sex, age, and weight (week 0).

The IBM SPSS® version 22 was used for the statistical analyses [74].

### Ethics

The study was registered in ClinicalTrials.gov (NCT04366609). The study was approved by the Danish Data Protection Agency (P-2019-851). The Local Committee on Health Research Ethics was consulted (J.nr. H-B-2009-026) in accordance with national guidelines and the Declaration of Helsinki and the study was evaluated not be within their jurisdiction due study design as an observational study. Participation was voluntary and data was kept confidential. The participants could withdraw their consent at any time without having to give reasons and with no consequences for their further treatment options.

## Results

### Baseline characteristics of included and excluded patients

Among the 542 patients screened for eligibility (Fig 1), a total of 207 patients (mean age 9.6 SD 1.5), 156 boys (75.4%)) were included in the study (Table 1). Among those with a verified ADHD diagnosis ($n$ = 418), a total of 211 patients were excluded. The main reason for exclusion were that the parents did not consent to MPH treatment of their children ($n$ = 95, 45.0%), a delayed decision to initiate MPH treatment, or no clinical indication for MPH treatment as judged by the clinician. Of the included patients, 187 (90.3%) patients (mean age 9.6 (SD 1.5), 140 boys (74.9%)) completed the 12 weeks study. There were relatively more males ($n$ = 177, 83.9%) among the excluded patients than included in the study ($p$ = 0.030) (Table 1).

The comparison of included versus excluded patients furthermore showed that the included patients had significantly higher ADHD-RS-P scores, and at the same time, significantly lower ADHD-RS-T scores compared with those excluded (S3 Table). There were no other significant differences in ADHD symptoms between included and excluded patients or between the included patients and the subgroup of patients excluded solely because their parents refused medicine ($n$ = 95). This group of patients who were excluded solely because of their parents' refusals were characterized by lower ADHD-RS-P scores compared to patients excluded due to other reasons ($n$ = 116), 10.9 (SD 5.3), and 13.9 (SD 6.3), respectively, ($p < 0.001$) (for further baseline characteristics see S4 Table). Only five children were excluded because of parents' refusals to let them participate in the study.

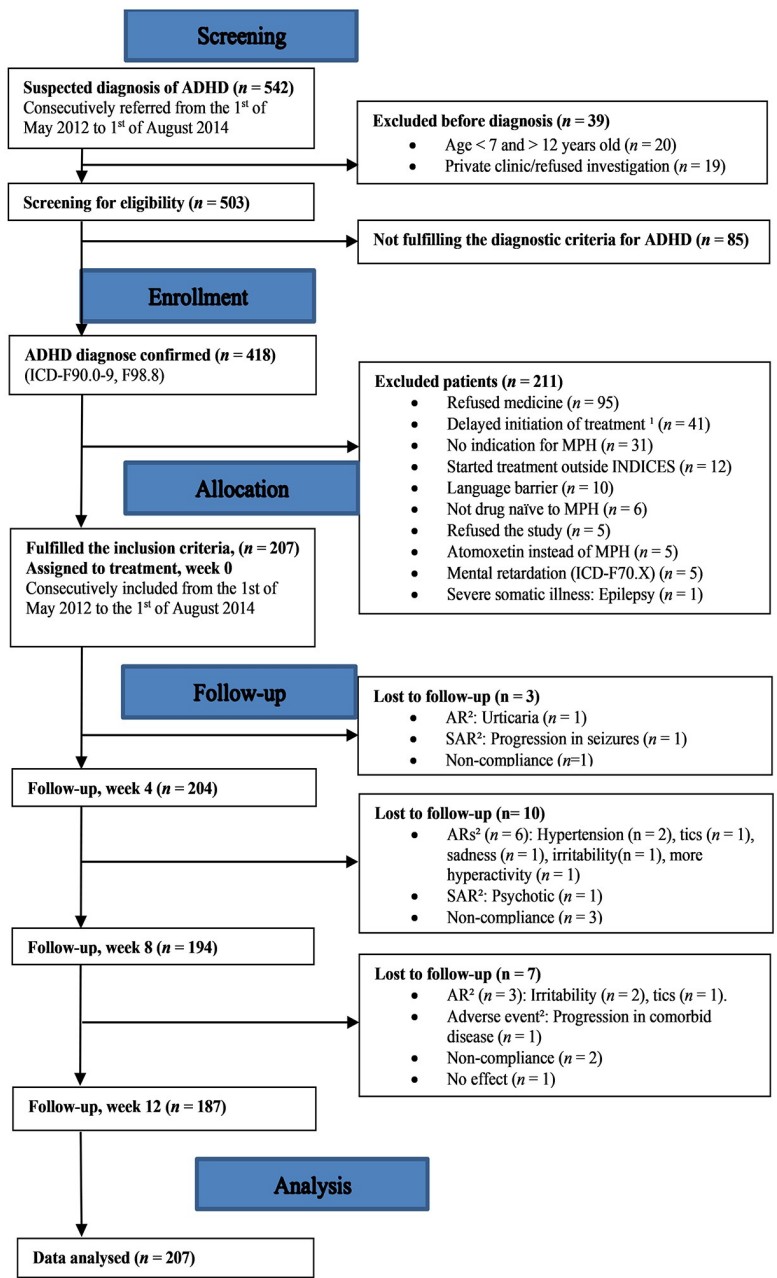

**Fig 1. TREND, flow diagram of inclusion.** ICD-10 = International Classification of Diseases and Related Health Problems, ADHD = Attention-deficit/hyperactivity disorder, MPH = Methylphenidate, INDICES = INDIvidualised drug therapy based on pharmacogenetics: focus on carboxylesterase 1, AR = Adverse reaction, SAR = Serious adverse reaction. [1] = Treatment initiated after the study inclusion period was terminated. [2] = Clinician decision of patient's discontinuation of treatment with MPH due to adverse events, ARs, and SARs.

## The treatment response during the first 12 weeks after initiation of IR-MPH treatment

**The reasons for discontinuation of IR-MPH treatment.** The clinicians decided when to discontinue medication with IR-MPH due to ARs or SARs ($n = 12$, 5.8%). Of the patients who discontinued the treatment with IR-MPH, one experienced onset of psychotic symptoms

**Table 1. Characteristics of included and excluded patients at study entry.**

| | Included (*n* = 207) | Excluded (*n* = 211) | Excluded, refused medicine[1] (*n* = 95) |
|---|---|---|---|
| Boys, *n* (%) | 156 (75.4) | 177 (83.9) | 75 (78.9) |
| Age, years, mean (SD) | 9.6 (1.5) | 9,9 (1.5) | 9.6 (1.6) |
| Age, 7–9 years, *n* (%) | 133 (64.3) | 119 (56.4) | 56 (58.9) |
| Age, 10–12 years, *n* (%) | 74 (35.7) | 92 (43.6) | 39 (41.1) |
| **ADHD diagnoses (ICD-10)** | | | |
| Disturbance of activity and attention, hyperkinetic disorder other, hyperkinetic disorder unspecified. (F 90.0, F 90.8, F 90.9) *n* (%) | 172 (83.1) | 163 (77.3) | 74 (77.9) |
| Hyperkinetic conduct disorder (F 90.1) *n* (%) | 12 (5.8) | 14 (6.6) | 5 (5.3) |
| Attention deficit disorder without hyperactivity (F 98.8,) *n* (%) | 23 (11.1) | 34 (16.1) | 16 (16.8) |
| **Comorbidity (ICD-10)** | | | |
| **Tic disorders.** (F95.X) *n* (%) | 17 (8.2) | 14 (6.6) | 4 (4.2) |
| **Externalizing disorders.** Conduct disorders, mixed disorders of conduct and emotions (F 91.X, F 92.X) *n* (%) | 14 (6.8) | 9 (4.3) | 4 (4.2) |
| **Specific developmental disorders.** (F81.X-83.X, F88.X) *n* (%) | 46 (22.2) | 46 (21.8) | 26 (27.4) |
| **Cognitive deficits.** Inferioritas intellectualis, mental retardation (R41.8[2], F70-79) *n* (%) | 56 (27.1) | 62 (29.4) | 27 (28.4) |
| **Encopresis and/or enuresis.** (F98.0–98.1) *n* (%) | 24 (11.6) | 18 (8.5) | 8 (8.4) |
| **Autism spectrum disorders.** (F84.X) *n* (%) | 26 (12.6) | 40 (19.0) | 15 (15.9) |
| **Attachment disorders.** (F94.X) *n* (%) | 3 (1.4) | 7 (3.3) | 1 (1.1) |
| **Emotional disorders.** Depressive episode, persistent mood-, obsessive-compulsive-, reaction to severe stress, and adjustment disorders, emotional disorders with onset specific to childhood. (F 32.X, F 34.X, F 42.X, F 43.X, F 93.X) *n* (%) | 27 (13.0) | 23 (10.9) | 10 (10.5) |
| **Number of comorbid psychiatric diagnoses** | | | |
| No comorbidity, *n* (%) | 72 (34.8) | 64 (30.3) | 29 (30.5) |
| 1 comorbid diagnosis, *n* (%) | 78 (37.7) | 85 (40.3) | 39 (41.1) |
| 2 comorbid diagnoses, *n* (%) | 39 (18.8) | 52 (24.6) | 25 (26.3) |
| 3 comorbid diagnoses, *n* (%) | 15 (7.2) | 10 (4.7) | 2 (2.1) |
| 4 comorbid diagnoses, *n* (%) | 3 (1.4) | 0 (0.0) | 0 (0.0) |

ICD-10 = International Classification of Diseases and Related Health Problems, ADHD = Attention-deficit/hyperactivity disorder.

[1] Patients, who refused medicine, were a subgroup of excluded patients.

[2] R41.8 referred to diagnose from the diagnostic conference and/or from the intelligence test (WISC) (IQ = 70–85) depending on data access.

(0.5%) and another patient had progression in seizures (0.5%), both were classified as SARs (1.0%). A total of 10 patients (4.8%) discontinued the treatment because of ARs: three patients (1.4%) because of irritability, two patients (1.0%) because of hypertension (increase in a systolic blood pressure and/or diastolic blood pressure to a level corresponding to the 95–99 percentile for age and gender), two patients (1.0%) because of tics, one patient (0.5%) because of sadness, one patient (0.5%) because of hyperactivity, and one patient (0.5%) because of urticarial reaction. Six of these 12 patients had ARs or SARs that were detected by the weekly monitoring with BSSERS-C, whereas the remaining ARs were reported spontaneously.

One other patient (0.5%) reported psychotic symptoms during treatment, and a thorough clinical evaluation established that intermittent psychotic symptoms had been present for several years before initiation of MPH treatment. The medication was discontinued, and the relapse of psychotic symptoms was classified as an adverse event. Furthermore, six patients (2.9%) were non-compliant/non-adherent and discontinued their treatment and one patient (0.5%) due to no symptom reduction.

**ADHD core symptoms, conduct problems, daily and social functioning.** A total of 137 (73.2%) patients obtained Nor/Bnor (70 (37.4%) patients obtained Nor) on the Inattention subscale (ADHD-RS-C) at week 12 (Nor and Bnor are separated in S5 Table). Of these patients, 111 (81.0%) were boys and 89 (65.0%) were 7 to 9 years of age. A total of 157 (84.0%) patients obtained Nor/Bnor (95 (58,8% patients obtained Nor) on the Hyperactivity-Impulsivity subscale (ADHD-RS-C) at week 12. Of these patients, 115 (55.6%) were boys and 103 (65.6%) were 7 to 9 years of age. In summary, 19 (10.1%) patients were not Nor/Bnor on any of the subscales at week 12, 126 (67.4%) patients were Nor/Bnor on both subscales of ADHD-RS-C at week 12, and 42 (22.5%) patients were Nor/Bnor on either the Inattention or the Hyperactivity-Impulsivity subscale of ADHD-RS-C at week 12.

Changes in sum scores from week 0 to week 12 showed a mean significant reduction and an effect size between 0.3 to 2.7 of severity of inattention symptoms and hyperactivity-impulsivity symptoms rated by clinicians, parents, and teachers (ADHD-RS-C, ADHD-RS-P, and ADHD-RS-T, Table 2). Reduction of conduct problems was statistically significant in parents' ratings ($p < 0.001$) but not in teachers' ratings ($p = 0.293$).

The mean percent reductions of scores on the clinician-rated ADHD-RS-C subscale from week 0 to 12 were 52.0% on Inattention and 56.0% on Hyperactivity-Impulsivity, and the mean percentage reductions on the parent-rated ADHD-RS-P subscales were 48.1% on Inattention, 45.0% on Hyperactivity-Impulsivity, and 50.7% on Conduct problems (S6 Table).

Patients significantly improved on the CGI-S scale from week 0 to week 12. A total of 67 (35.8%) patients were rated with a score of 1 or 2 (*normal to mildly ill*) on CGI-S after 12 weeks of treatment ($p < 0.001$, the mean difference between week 0 and 12 was 0.4 (SD 0.5, 95%CI 0.3 to 0.4). The mean score on CGI-S was 5.3 (SD 1.0) in week 0 and 3.0 (SD 1.1) after 12 weeks of treatment ($p < 0.001$, M difference 2.3, SD 1.0, 95%CI 2.2 to 2.5). A total of 171 patients (91.5%) had a score of 1 or 2 (*very much* or *much improved*) on CGI-I after 12 weeks of treatments. The mean score on CGI-I at week 12 was 1.8 (SD 0.6) (Table 2).

After 12 weeks of treatment, the patients' daily and social functioning were improved on WFIRS-P, which showed a significant reduction of symptoms of all six subscales from week 0 to week 12 (S7 Table). Mean subscale index scores of WFIRS-P at week 0 and 12 were ≤1, meaning that the overall level of functioning was mildly affected in each domain (range of index score 0–3). The largest mean reductions on the WFIRS-P were on the school subscale and the social life subscale with mean reductions of 0.4 (SD 0.4) and 0.4 (SD 0.5), respectively, and the smallest mean reductions on the subscales were on the daily life subscale and the risk behaviour subscale with mean reductions of 0.1 (SD 0.4) and 0.1 (SD 0.2). All reductions were statistically significant and had effect sizes between 0.2 to 1.6.

The four mean TOVA outcomes showed significant improvements and effect sizes between 0.6 to 1.1 in response time, variability of response time, omission errors, and commission errors ($n = 115$) from week 0 to week 12. Compared to week 0, at week 12 there were improvements in standard deviations of 1.0 on response time, 1.4 on variability of response time, 0.9 on omission errors, and 0.7 on commission errors (S7 Table). The mean time from midday IR-MPH dose to TOVA test was 108 minutes (range 9 to 193), and the mean midday dose was 0.42 mg IR-MPH per kg (SD 0.13) ($n = 106$) at week 12.

**Attrition during the study.** There were no differences between the patients who completed the study ($n = 187$) and those who discontinued IR-MPH treatment due to ARs ($n = 12$) at entry (week 0) in mean sum scores of Inattention and Hyperactivity-Impulsivity (ADHD-RS-C) (19.9 (SD 3.7) and 18.0 (SD 5.8) vs. 20.3 (SD 3.2) and 19.7 (SD 5.2), $p = 0.742$ and $p = 0.315$) and in mean sum score of ARs (BSSERS-C) (17.6 (SD 10.8) vs. 17.9 (SD 7.8), $p = 0.928$).

**Table 2. Clinical follow-up from week 0 to week 12 ($n$ = 187).**

| ADHD rating scales | Week 0 M (SD), n (%) | Week 12 M (SD), n (%) | Week 0 versus week 12 | | | | Cohen's d |
|---|---|---|---|---|---|---|---|
| | | | M dif. (SD) | 95% CI | t (df) | p-value | |
| **Parent** | | | | | | | |
| **Inattention** | 17.4[1] (5.0) | 9.2[1] (4.2) | 8.2 (5.1) | (7.5, 9.0) | 21.3 (173) | < 0.001 | 1.8 |
| **Hyperactivity-Impulsivity** | 15.6[2] (6.0) | 8.6[2] (4.7) | 7.0 (5.7) | (6.2, 7.9) | 16.2 (170) | < 0.001 | 1.2 |
| **Inattention and Hyperactivity-Impulsivity** | 33.2[3] (9.4) | 17.7[3] (7.9) | 15.5 (9.4) | (14.1, 17.0) | 21.1 (162) | < 0.001 | 1.6 |
| **Conduct problems** | 10.7[4] (5.9) | 5.3[4] (4.1) | 5.4 (5.1) | (4.7, 6.2) | 14.2 (178) | < 0.001 | 1.1 |
| **Teacher** | | | | | | | |
| **Inattention** | 16.9[5] (4.7) | 11.2[5] (5.1) | 5.6 (5.1) | (4.7, 6.5) | 12.4 (126) | 0.001 | 1.1 |
| **Hyperactivity-Impulsivity** | 9.1[6] (5.7) | 7.7[6] (4.9) | 1.4 (5.2) | (0.5, 2.3) | 3.0 (124) | 0.004 | 0.3 |
| **Inattention and Hyperactivity-Impulsivity** | 25.9[7] (9.0) | 18.6[7] (8.4) | 7.2 (8.9) | (5.6, 8.9) | 8.8 (117) | < 0.001 | 0.8 |
| **Conduct problems** | 4.8[8] (4.7) | 4.4[8] (4.1) | 0.4 (4.4) | (-0.4, 1.2) | 1.1 (127) | 0.293 | 0.1 |
| **Clinician** | | | | | | | |
| **Inattention** | 19.9 (3.7) | 9.6 (3.7) | 10.3 (4.0) | (9.7, 10.9) | 35.2 (186) | < 0.001 | 2.6 |
| **Hyperactivity-Impulsivity** | 18.0 (5.8) | 7.9 (4.0) | 10.1 (5.1) | (9.4, 10.8) | 27.3 (186) | < 0.001 | 2.0 |
| **Inattention and Hyperactivity-Impulsivity** | 37.9 (7.8) | 17.5 (6.8) | 20.4 (7.7) | (19.3, 21.5) | 36.5 (186) | < 0.001 | 2.7 |
| **CGI-S** | 5.3 (1.0) | 3.0 (1.1) | 2.3 (1.0) | (2.2, 2.5) | 31.5 (186) | < 0.001 | 2.2 |
| **CGI-S ≤ 2** | 0 (0.0%) | n = 67 (35.8%) | 0.4 (0.5) | (0.3, 0.4) | 10.2 (186) | < 0.001 | |
| **CGI-I** | - | 1.8 (0.6) | - | - | - | - | |
| **CGI-I ≤ 2** | - | n = 171 (91.5%) | - | - | - | - | |
| **Weight,** kg | 36.3[9] (11.0) | 35.4[9] (11.1) | 0.9 (1.5) | (0.7, 1.15) | 8.3 (183) | < 0.001 | 0.6 |
| **Height,** cm | 140.1[9] (10.8) | 141.3[9] (10.7) | -1.2 (1.1) | (-1.4, -1.0) | -14.8 (183) | < 0.001 | 1.1 |
| **Diastolic blood pressure,** mmHg | 67.1[10] (7.1) | 66.4[10] (7.4) | 0.7 (9.4) | (-0.7, 2.0) | 0.9 (182) | 0.344 | 0.1 |
| **Systolic blood pressure,** mmHg | 102.2[10] (8.9) | 101.8[10] (9.7) | 0.3 (9.9) | (-1.2, 1.7) | 0.4 (182) | 0.693 | 0.0 |
| **Heart rate,** bpm | 79.2[11] (10.9) | 78.2[11] (11.9) | 1.0 (15.1) | (-1.2, 3.3) | 0.9 (180) | 0.341 | 0.1 |
| **BSSERS-C** | 17.3[9] (10.5) | 14.5[9] (9.0) | 2.8 (8.6) | (1.5, -4.0) | 4.4 (183) | < 0.001 | 0.6 |
| **Reduced appetite** | 0.7[12] (1.3) | 1.9[12] (1.5) | -1.2 (1.8) | (-1.5, 1.0) | -9.2 (185) | < 0.001 | 0.7 |
| **Methylphenidate** mg/kg/day | 0.3 (0.1) (initial dose) | 1.0 (0.3) | -0.7 (0.3) | (-0.7, -0.7) | -34.2 (186) | < 0.001 | 2.4 |

Paired t-test between outcomes of week 0 and week 12. M = mean, M dif. = Mean difference, SD dif. = Standard deviation difference, $n$ = number. Number of participants with observed outcome data: [1] $n$ = 174, [2] $n$ = 171, [3] $n$ = 163, [4] $n$ = 179, [5] $n$ = 127, [6] $n$ = 125, [7] $n$ = 118, [8] $n$ = 128, [9] $n$ = 184, [10] $n$ = 183, $n$ [11] = 181, [12] $n$ = 186. ADHD-Rating Scale (ADHD-RS, DuPaul). Inattention subscale: 9 items, [range 0–27]. Hyperactivity-Impulsivity subscale: 9 items [range 0–27]. Conduct problems subscale: 8 items, [range 0–24]. Rated by clinician, parent, or teacher.

Clinical Global Impression Severity (CGI-S): 1 item, [range 1–7]. CGI-S, 1 = *Not ill at all*. CGI-S, 2 = *Borderline ill*.

Clinical Global Impression Improvement (CGI-I): 1 item, [range 1–7]. CGI-I, 1 = *Very much improved*. CGI-I, 2 = *Much improved*.

Barkley's Stimulant Side Effects Rating Scale, clinician rated (BSSERS-C). Whole scale: 17 items, [range 0–153]. Reduced appetite: single item, [range 0–9].

**Adverse reactions.** Overall the mean sum problem score of the BSSERS-C rating scale significantly declined over 12 weeks of treatment (pre-post mean difference 2.8 (SD 8.6), $p < 0.001$) (Table 2, single-item ARs of BSSERS-C in S8 Table). Reduced appetite was the only BSSERS-C-rated AR problem score that increased significantly over time, whereas the other BSSERS-C-rated AR problem scores were either stable or decreased over time (e.g., the largest decrease on any single AR problem score was found for euphoria with a mean decrease of 0.7 (SD 1.4) $p < 0.001$, 95%CI 0.5 to 0.9). There was no significant increase in mean blood pressure, and no change in heart rate from week 0 to week 12. There was a significant increase in children's height with a mean of 1.2 cm (SD 1.1) despite a mean loss of weight of 0.9 kg (SD 1.5) (Table 2).

**Nonresponders.** Thirty-one patients (15.0%) were nonresponders; of these, 12 (5.8%) patients discontinued IR-MPH treatment because of ARs/SARs, and 19 patients (9.1%) were not Nor/Bnor on any of the ADHD-RS-C subscales. Responders (*n* = 168) and nonresponders did not differ with respect to age, sex, and comorbidity (S9 Table). Nonresponders were characterised by having more severe ADHD and global symptoms than responders at week 0 (mean sum scores of Hyperactivity-Impulsivity were 21.5 (SD 4.4) vs. 17.5 (SD 5.7), $p \leq 0.001$, and mean CGI-S scores were 5.9 (SD 0.8) vs. 5.2 (SD 0.9), $p \leq 0.001$, and a higher mean score of problems on BSSERS-C (21.3 (SD 11.3) vs. 17.0 (SD 10.4), $p = 0.034$) at week 0.

## Modelling the response and the predictors of response during the first 12 weeks of treatment

**Course of inattention.** Among the investigated clinical characteristics, sex, age, and time (number of weeks in treatment) were significantly associated with the course of inattention symptoms, rated on the Inattention subscale (ADHD-RS-C, range 0–27), when analysed in the linear mixed effect model (Table 3, graphs of Inattention Fig 2A). The estimated effects of time corresponded to a mean reduction of 0.8 of the Inattention sum score per week in subgroups of sex and age throughout the treatment period. Girls aged 7 to 9 years had a lower estimated mean score of Inattention symptoms in week 0 (score 17.5) than the group of girls aged 10 to 12 years (score 18.6) in week 0. Boys aged 7 to 9 years had a higher estimated mean score of Inattention symptoms in week 0 (score 19.4) than boys aged 10 to 12 years (score 18.2) in week 0. None of the investigated variables (age, sex, intelligence level, conduct disorder, and comorbidity) had a significant interaction with time, and the differences in estimated mean scores in week 12 between sex and age were the same as in week 0.

**Course of hyperactivity-impulsivity.** In the linear mixed effect model for repeated measures, only age and time (number of weeks in treatment) were significantly associated with the course of hyperactivity-impulsivity symptoms, rated on the Hyperactivity-Impulsivity subscale (ADHD-RS-C, range 0–27) (Table 3, graphs of Hyperactivity-Impulsivity Fig 2B). Patients aged 7 to 9 years had an estimated baseline score of 16.0 and an estimated decrease in symptoms of 0.8 per week. Patients aged 10 to 12 years had an estimated baseline score of 13.3 and estimated decrease in symptoms of 0.6 per week. In week 12, patients aged 7 to 9 years had a score of 6.8 and patients aged 10 to 12 years had an estimated score of 6.3 (i.e., the scores from the two age groups of patients ended up at about the same level).

**Normalisation or borderline normalisation of the clinician-rated ADHD core symptoms.** Female sex (HR = 0.64, 95% CI 0.54 to 0.75) and a diagnosis of *inferioritas intellectualis* (HR = 0.74, 95%CI 0.63 to 0.86) were significant predictors of a lower chance of Nor/Bnor on the Inattention subscale (ADHD-RS-C) (Table 4).

Young age (7–9 years) (HR = 0.88, 95%CI 0.71 to 0.96) and *inferioritas intellectualis* (HR = 0.82, 95%CI 0.78 to 0.99) significantly predicted lower chance of Nor/Bnor on the Hyperactivity-Impulsivity subscale (ADHD-RS-C). Comorbidity (HR 1.17, 95%CI 1.02 to 1.35) significantly improved the chance of Nor/Bnor on the Hyperactivity-Impulsivity subscale (ADHD-RS-C) (Table 4). Post hoc analysis did not show any significant differences in sum scores of Inattention or Hyperactivity-Impulsivity subscale (ADHD-RS-C) in week 0 between patients with none or one psychiatric comorbid diagnosis and those with two or more psychiatric comorbid diagnoses.

**Adverse reactions.** The mean reduction in the BSSERS-C problem score was 0.3 per week. Among the investigated clinical characteristics, only CGI-S measured in week 0 and time (number of weeks in treatment) were significantly associated with the course of ARs in the linear mixed effect model for repeated measures (BSSERS-C, range 0–153) (Table 3, graphs

**Table 3. Baseline characteristics as predictors for symptom reductions and adverse reactions (*n* = 207).**

| Linear mixed model | | | |
|---|---|---|---|
| **Inattention** | **Estimate** | **95% CI** | **p-value** |
| Intercept | 18.21 | (17.37, 19.05) | < 0.001 |
| Week | - | - | < 0.001 |
| Week | -0.77 | (-0.81, -0.72) | < 0.001 |
| Age | - | - | 0.062 |
| 10–12 years (ref.) | - | - | - |
| 7–9 years | 0.38 | (-0.66, 1.30) | 0.521 |
| Sex | - | - | 0.794 |
| Boys (ref.) | - | - | - |
| Girls | 1.38 | (-0.19, 2.94) | 0.084 |
| Age*Sex | - | - | 0.013 |
| Boys*10–12 years | - | - | - |
| Boys*7–9 years | - | - | - |
| Girls*10–12 years | - | - | - |
| Girls*7–9 years | -2.50 | (-4.46, -0.54) | 0.013 |
| **Hyperactivity-Impulsivity** | **Estimate** | **95% CI** | **p-value** |
| Intercept | 13.30 | (12.13, 14.47) | < 0.001 |
| Week | - | - | < 0.001 |
| Week | -0.58 | (-0.66, -0.50) | < 0.001 |
| Age | - | - | < 0.001 |
| 10–12 years (ref.) | - | - | - |
| 7–9 years | 2.65 | (1.19, 4.11) | < 0.001 |
| Week*age | - | - | < 0.001 |
| Week*10–12 years (ref.) | - | - | - |
| Week*7–9 years | -0.18 | (-0.28, -0.08) | < 0.001 |
| **Adverse reactions** | **Estimate** | **95% CI** | **p-value** |
| Intercept | 21.66 | (19.84, 23.49) | < 0.001 |
| Week | - | - | < 0.001 |
| Week | -0.28 | (-0.37, -0.19) | < 0.001 |
| CGI-S, week 0 | - | - | < 0.001 |
| CGI-S, 6–7 (ref.) | - | - | - |
| CGI-S, 5 | -4.53 | (-7.00, -2.06) | < 0.001 |
| CGI-S, 4 | -4.30 | (-7.77, -0.84) | 0.015 |
| CGI-S, 1–3 | -8.25 | (-14.15, -2.35) | 0.006 |

| General Estimating Equation | | | | | |
|---|---|---|---|---|---|
| **Reduced appetite** | **Estimate** | **95% CI** | **p-value** | **OR** | **95% CI** |
| Intercept | 1.90 | (1.44, 2.37) | < 0.001 | 6.71 | (4.21, 10.71) |
| Week | - | - | < 0.001 | | |
| Week | -0.04 | (-0.08, 0.00) | 0.069 | 0.96 | (0.93, 1,00) |
| CGI-S, week 0 | - | - | < 0.001 | | |
| CGI-S 6–7 (ref.) | - | - | - | | |
| CGI-S 1–5 | 0.88 | (0.35, 1.41) | < 0.001 | 2.41 | (1,42, 4,08) |

**Linear mixed model for repeated measures** of sum scores of Inattention, Hyperactivity-Impulsivity, and adverse reactions. ADHD-Rating Scale, clinician rated. (ADHD-RS-C, DuPaul). Inattention: 9 items, [range 0–27]. Hyperactivity-Impulsivity: 9 items, [range 0–27]. Barkley's Stimulant Side Effects Rating Scale, clinician rated (BSSERS-C). Whole scale: 17 items, [range 0–153]. Only significant values ($p < 0.05$) of the explanatory variables as main factors and/or interactions are listed in the table. Explanatory variables; continuous variable: time (defined as 13 weeks from week 0 to 12) and categorical variables: sex, age, Clinical Global Impression Severity (CGI-S) in week 0. CGI-S, [range 1–7], divided into four groups: CGI-S 1–3 = *not ill at all*, *borderline ill*, *mildly ill*, CGI-S 4 = *moderately ill*, CGI-S 5 = *markedly ill*, CGI-S 6–7 = *severely ill*, *among the most extreme ill*.

**General Estimating Equation for repeated measures** of reduced appetite. BSSERS-C, single item, reduced appetite, [range 0–9]. No or mild reduced appetite 0–3, moderate to server reduced appetite 4–9. Only significant values ($p < 0.05$) of the explanatory variables as main factors and/or interactions are listed in the table. Explanatory variables; continuous variable: time (defined as 13 weeks from week 0 to 12) and categorical variables: CGI-S in week 0 divided into CGI-S 1–5 and CGI-S 6–7.

Estimate refers to the sum score of the scale (ADHD-RS-C or BSSERS-RS).

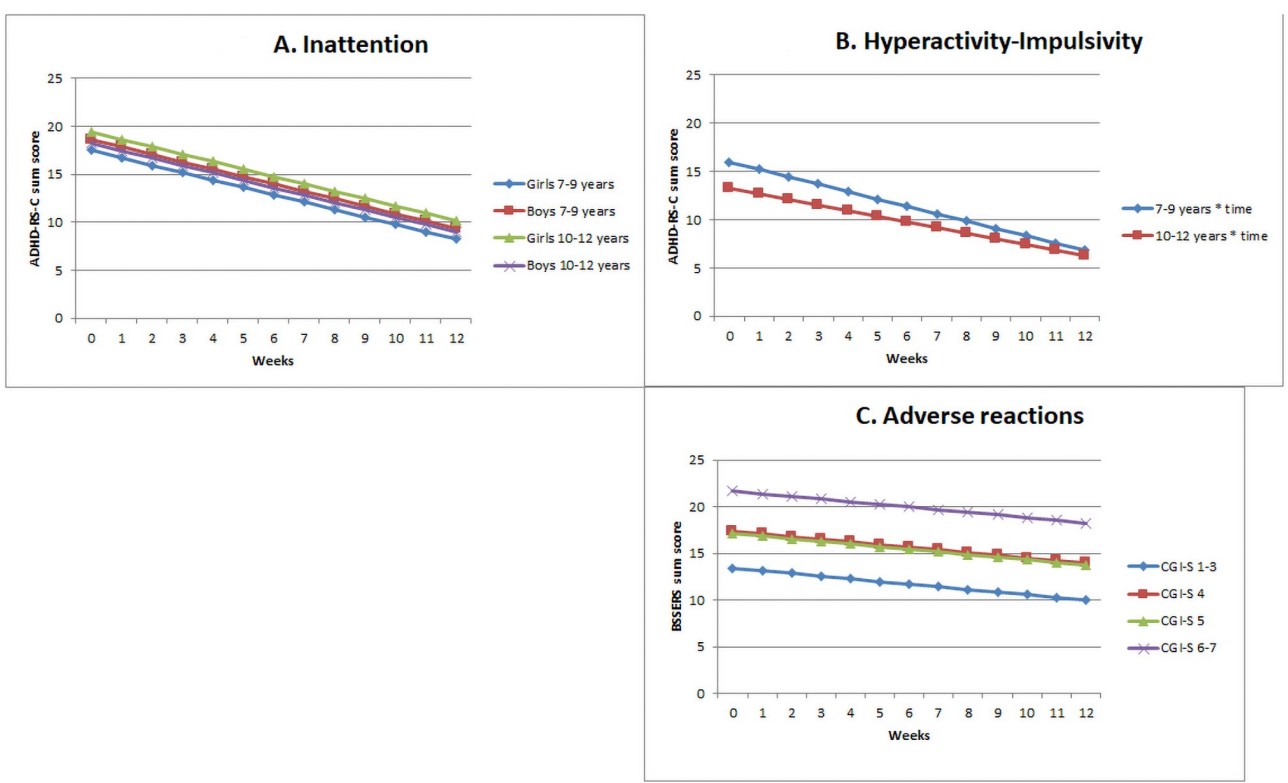

**Fig 2. Examples of graphs for estimations sum scores of inattention, hyperactivity-impulsivity, and adverse reactions throughout the 12 weeks as estimated by the linear mixed models, *n* = 207.** Clinician rated ADHD-Rating-Scale (ADHD-RS-C); Inattention subscale, 9 items [range 0–27] and Hyperactivity-Impulsivity subscale, 9 items [range 0–27]. Clinician rated Barkley's Stimulant Side Effect Rating Scale (BSSERS-C), 17 items, [range 0–153]. Explanatory variables: Sex, age (7 to 9 years and 10 to 12 years), functional impairments in week 0, and Clinical Global Impression Severity (CGI-S). CGI-S devided into: 1–3 = *Not ill, borderline ill, mildly ill*. CGI-S 4 = *Moderately ill*, CGI-S 5 = *Markedly ill*. CGI-S 6–7 = *Severly ill, among the most extreme ill patients*.

of ARs Fig 2C). Patients rated *severely ill* and *most extremely ill* (CGI-S 6–7) in week 0 had a more severe BSSERS-C problem score (mean score 21.7) before initiation of treatment than patients with a lower-rated CGI-S score in week 0. Patients who were rated *moderately ill* (CGI-S 5) at baseline had an estimated mean BSSERS-C problem score of 17.2, patients rated *markedly ill* on CGI-S (CGI-S 4) had an estimated mean BSSERS-C problem score of 17.4, and patients rated *not ill at all*, *borderline ill*, or *mildly ill* (CGI-S 1–3) had an estimated mean BSSERS-C problem score of 13.4 before initiation of treatment. None of the investigated variables (age, sex, intelligence level, conduct disorder, comorbidity, time, ADHD diagnosis, and CGI-S in week 0) had a significant interaction with time.

**Reduced appetite.** Of the investigated AR items on BSSERS-C, reduced appetite was the only item that significantly increased in problem score over the 12-week treatment period. Reduced appetite was post hoc defined as a problem score of 4–9 on BSSERS-C. Among the investigated clinical characteristics in week 0, only CGI-S was significantly associated with the course of reduced appetite in the model of GEE for repeated measures (BSSERS-C, single item, 0–3 = no reduced appetite, 4–9 = reduced appetite) (Table 3). *Severely ill* and *most extremely ill* rated patients (CGI-S 6–7) had a significantly higher risk of reduced appetite compared with patients rated not ill to moderately ill (CGI-S 1–5) in week 0 and throughout the whole treatment period (OR = 2.41, 95%CI 1.42 to 4.08). The probability of reduced appetite in week 0 was 13.0% for patients rated severely ill and most extremely ill and 5.8% for patients not ill to

**Table 4. Baseline characteristics as predictors for the chance of normalisation or borderline normalisation of the clinician rated ADHD core symptoms score ($n$ = 207).**

| Predictors for the chance of normalisation or borderline normalisation of ADHD symptoms | | | |
|---|---|---|---|
| **Inattention** | **HR** | **95% CI** | ***p*-value** |
| Sex (girls) | 0.64 | (0.54, 0.75) | < 0.001 |
| Inferioritas intellectuals | 0.74 | (0.63, 0.86) | < 0.001 |
| **Hyperactivity-Impulsivity** | **HR** | **95% CI** | ***p*-value** |
| Age (7–9 years) | 0.88 | (0.71, 0.96) | 0.031 |
| Inferioritas intellectuals | 0.82 | (0.78, 0.99) | 0.011 |
| Comorbidity | 1.17 | (1.02, 1.35) | 0.030 |

**Cox regressions** of time to first normalisation (zero standard deviation) or borderline normalisation (one standard deviation) of ADHD core symptoms due to Danish norms of sex and age on ADHD-Rating Scale, clinician rated (ADHD-RS-C, DuPaul). Inattention subscale: 9 items, range 0–27. Hyperactivity-Impulsivity subscale: 9 items, range 0–27. Only last model of cox regression with backward elimination are listed in the table. HR = Hazard ratio. Independent predictors: Sex, age, comorbidity (≤ 2 comorbid diagnosis), and intelligence level (*inferioritas intellectuals*, ICD10-R41.8) from the diagnostic conference and/or from the intelligence test (WISC) (IQ = 70–85) depending on data access).

*moderately ill.* The proportion of patients with reduced appetite did not change significantly over time (weeks, $p$ = 0.069).

**Baseline characteristics as predictors for end-dose of IR-MPH.** The mean end-dose of IR-MPH was 1.0 mg/kg/day (SD 0.3, Table 2). Higher CGI-S scores at week 0 predicted a higher end-dose of IR-MPH. CGI-S at week 0 and age explained together 12.7% ($p$ < 0.001) of the variance of the end-dose IR-MPH. The estimated effect of being "among the most extremely ill patients" in week 0 (CGI-S 7) corresponded to an end-dose of IR-MPH 1.19 mg/kg/day, whereas being rated "mildly ill" in week 0 (CGI-S 3) corresponded to an end-dose of IR-MPH 0.85 mg/kg/day (Table 5). Young age predicted a lower end-dose of IR-MPH. Patients aged 7 to 9 years were estimated to receive a 0.13 mg/kg/day lower end-dose of IR-MPH than patients aged 10–12 years.

## Discussion

This study is a prospective, uncontrolled, ecologically valid study aimed at exploring the course and outcome of a group of patients treated with IR-MPH treatment in a routine child and adolescent clinic in a real-world setting [29]. This heterogeneous group of patients with substantial psychiatric comorbidity, including autism spectrum disorders and *inferioritas intellectualis*, were offered an individually titrated optimal dosing of IR-MPH based on weekly assessments

**Table 5. Baseline characteristics as predictors for end-dose of IR-MPH ($n$ = 187).**

| Predictors for end-dose of IR-MPH dose/mg/kg/day | | | | | | |
|---|---|---|---|---|---|---|
| | **b** | **std Error** | **Beta** | **t** | ***p*-value** | **95% CI of estimate** |
| Constant | 0.56 | 0.12 | | 4.84 | < 0.001 | (0.33, 0.79) |
| CGI-S, week 0 | 0.09 | 0.02 | 0.29 | 4.32 | < 0.001 | (0.05, 0.13) |
| Age (7–9 years) | -0.13 | 0.04 | -0.16 | -3.05 | 0.003 | (-0.21, -0.05) |

**Linear regression** for end-dose of Immediate Release Methylphenidate (IR-MPH).

Explaining variables are measured in week 0. Clinical Global Impression Severity (CGI-S) in week 0, [range 1–7].

Model Summary: R = 0.37, R Square = 0.14, Adjusted R Square = 0.13, Std. Error of the Estimate = 0.27.

of ADHD core symptoms and ARs. Children with ADHD displayed statistically and clinically significant improvements of these core symptoms across informants, global level of symptoms, and parent-reported impairments of daily life functioning. The clinical findings were supported by an objective performance based measure of the patients' sustained attention by TOVA, which also showed significant improvements after 12 weeks of treatment with IR-MPH. Altogether, 15% were classified as nonresponders, and compared with the responders, they were characterised by more severe symptoms of hyperactivity-impulsivity and global impairment before initiation of treatment. Interestingly, the mental and physical complaint, measured as potential ARs, scored highest before initiation of treatment and dropped significantly during the up-titration of IR-MPH, suggesting that the complaints reflected general health and mental health problems which were related to the ADHD illness rather than ARs to IR-MPH treatment. However, two types of well-known ARs to IR-MPH (reduced appetite and weight loss) increased during the period of IR-MPH treatment. To our knowledge this is the first naturalistic observational study that monitored ARs weekly in a 12-week period.

Results were mixed regarding potential predictors of outcome of the individually titrated IR-MPH-treatment, suggesting no need to modify the dosing of IR-MPH according to age, gender, IQ between 70 and 85, or psychiatric comorbidity. Although the chance of Nor/Bnor was reduced for children with *inferioritas intellectualis* and increased for children with psychiatric comorbidity in the Cox regression analyses, this pattern was not confirmed in the more powerful mixed models of the course of symptoms.

The individually titrated doses varied (range 0.3–2.0 mg/kg/day), and higher levels of clinical global impairments before initiation of treatment predicted higher end-doses, whereas being between 7 and 9 years old was associated with slightly lower end-doses. The mean end-dose of 1.0 mg/kg/day was within the clinical recommendations.

The pattern of higher parent-rated ADHD core symptoms and, at the same time, lower teacher-rated ADHD core symptoms at entry was seen among included children for whom the parents accepted IR-MPH-treatment, and the opposite pattern was seen among children excluded due to parents' refusals of IR-MPH-treatment. This is a strong reminder that parents' conceptions of child ADHD core symptoms may impact the clinical decision to offer IR-MPH treatment of children.

Outcomes of MPH treatment vary in the literature, which may reflect definitions of response criterions used, the choice of ADHD rating scales, the length of observation period and the type of informants. Compared to a systematic review of efficacy trials (stimulant medication) reporting symptom reductions in the range of 10 to 18 in absolute scores [24], this study found a mean symptom reduction of 20.4 (SD 7.7) on the ADHD-RS-C, suggesting clinical effects within the same range as reported in efficacy trials. Furthermore, the effect sizes from 0.8 to 2.7 on ADHD core symptoms (a part from the teacher rated hyperactivity-impulsivity symptoms with an effect size of 0.3) were comparable to metanalyses with effect sizes from 0.54 to 1.78) for pre-post change in ADHD core symptoms after initiation of MPH treatment [14, 15], Direct comparisons of the present findings with results from other naturalistic observational studies with MPH naïve patients are only possible by comparing scores on the CGI-S and CGI-I (response rates defined as CGI-I $\leq$ 2 and CGI-S $\leq$ 2 at end of study). Kim *et al.* [32] found a higher CGI-S response rate (46.1%) after 12 weeks of treatment compared to the response rate in the present study (35.8%). Whereas two other studies found lower CGI-I response rates; 79.1% after 8 weeks of treatment [41] and 49.9% after 24 weeks of treatment [35] than the response rate of 91.5% in the present study (S2 Table and Table 2). This could indicate that, the present study included very ill patients, but the patients still had high response rates measured with CGI-I after 12 weeks of treatments compared to other studies.

The present study found a significant improvement of the level of daily and social functioning (total mean score of WFIRS-P) though the overall level of functioning in daily life rated by parents, was only mildly affected at study entry [60].

In line with findings in RCTs [64, 65] and naturalistic observational studies [35, 33], we found significant improvements of attention on all TOVA parameters. The variability of response time exhibited the largest improvement, indicating higher stability of attention during the whole test period.

Contrary to expectations, problem scores rated by the clinicians as potential ARs (BSSERS-C) were rated highest before medication and decreased during the treatment period (see S6 Table for single items), with the exception of appetite (BSSERS-C) that decreased along with a decrease in weight during treatment. The 17-item rating scale of BSSERS has been validated in a triple-blind placebo-controlled crossover study ($n$ = 83), in which parents and teachers rated ARs of MPH in children with ADHD [49]. In that study, BSSERS was found useful to measure reduced appetite, insomnia, headache, and stomachache (rated by parents), and staring, sadness, and anxiety (rated by teachers) during the treatment period, but the same ARs were also reported at lower levels during the placebo periods.

Many of the AR items rated on BSSERS-C may not be true ARs associated to IR-MPH but may reflect symptoms of ADHD or psychiatric comorbidity [75]. This may challenge the frequent use of BSSERS as a valid instrument for assessment of ARs associated with stimulant treatment [15, 33, 34, 39, 48]. Furthermore, only half of the patients who dropped out due to ARs experienced ARs detected on BSSERS-C in this study. There are few other validated AR rating scales, such as the Pittsburgh Side-Effects Rating Scale [76] and Subject's Treatment Emergent Symptom Scale [77], which are not very frequently used in research. When studies systematically use the ARs ratings scales, very few ARs increases during treatment with MPH and even decreases of "ARs" are found (*irritability*, *proneness to cry*, *anxiety*, *nail biting*, *euphoria*, and *sadness*) [75, 78, 79]. But the literature search revealed a general lack of systematic reporting of ARs, and several studies limited AR assessments to spontaneous reports. Reduced appetite and insomnia are the most frequent reported ARs in RCTs [15, 80] and in naturalistic observational studies [33, 34, 39]. The present study had a low discontinuation rate due to ARs (5.8%). Similar rates are found in other naturalistic observational studies (8.0% in [31] and 4.5% in [33]). Our findings are in line with a recent Cochrane review of non-randomized studies of children (5–18 years) with ADHD treated with MPH, which identified a discontinuations rate of 1.2% due to SARs and a discontinuations rate of 6.2% due ARs [81]. The discontinuation rates were low despite of relative high frequencies of reported ARs (insomnia (17.9%), headache (14.4%), abdominal pain (10.7%), and reduced appetite (6.2%)). In the present study, there was no increase in mean blood pressure during IR-MPH-treatment, which is in line with findings from another naturalistic observational study [36]. However, two of our patients discontinued treatment due to hypertension. Overall, the patients who completed the 12 weeks of treatment in the present study seemed to report fewer clinically significant ARs than patients in other naturalistic observational studies [33, 39] (S2 Table).

Children aged 7–9 years had a lower chance of normalisation or borderline normalisation of hyperactivity-impulsivity symptoms (Nor/Bnor of ADHD-RS-C) after 12 weeks of treatment than children aged 10–12 years, even though, children aged 7–9 years had a higher sum score of hyperactivity-impulsivity symptoms (ADHD-RS-C) in week 0 and had a higher weekly reduction of symptoms than children aged 10–12 years. In both cases, the differences were minor and hence of limited clinical significance. Also, the differential reduction of ADHD symptoms was explained by symptom severity differences at entry, whereas the end scores were similar for the two age groups.

The subgroup of girls aged 7–9 years had a lower sum score of inattention symptoms (ADHD-RS-C) than girls of age 10–12 years throughout the treatment period, but the weekly reduction of inattention symptoms was the same regardless of age and gender. The contradicting finding that girls had a lower chance of normalisation or borderline normalisation of inattention symptoms (ADHD-RS-C) compared to boys might be an artefact related to the gender specific cut-offs for normalisation based on a study conducted 10 years ago [55]. This point toward a need for new *age and gender* specific norms for ADHD core symptom scores.

IQ above 85 versus IQ below or equal to 85 but above 70 (*inferioritas intellectualis*) was statistically significantly associated with a better chance of Nor/Bnor of the inattention and hyperactivity-impulsivity symptoms, whereas the weekly reduction of sum scores of inattention and of hyperactivity-impulsivity showed no differences between IQ groups. These seemingly mixed results might indicate equally beneficial effects in both groups even though the chance of normalization is reduced when IQ is reduced.

Having two or more comorbid psychiatric diagnoses predicted a better chance of normalisation or borderline normalisation of hyperactivity-impulsivity symptoms (Nor/Bnor of ADHD-RS-C), but this finding contrasted with the statistical modelling of the course of hyperactivity-impulsivity symptoms, which showed no significant association with comorbidity.

Results from the Multimodal Treatment Attention Deficit Hyperactivity Disorder (MTA) study [82] showed that patients with an intelligence level above the population mean (IQ 100) had a better response to MPH treatment than patients with lower IQ. Other RCTs also found a normal IQ range is a predictor for a good treatment outcome [83–85], which also was found in one naturalistic observational study [86] but not in another [87]. Many different baseline characteristics have been sporadically identified as predictors for good treatment response (and sometimes opposite directed) in RCTs [82, 85] and naturalistic observational studies [30, 87]. The most consistent result of a predictor for a good treatment outcome might be intelligence within in the normal to upper range.It is well known that ARs of MPH are related to high doses of MPH as observed in RCTs [78, 88] and in a naturalistic observational study [34], but a review of 10 RCTs did not find the same association [80]. Severe global impairment (CGI-S 6–7) in week 0 predicted a higher mean problem score of ARs (BSSERS-C) throughout the whole treatment period including week 0, and severe global impairment (CGI-S 6–7) in week 0 was associated with a high end-dose of IR-MPH. Taken together, these findings indicate that the more severely ill patients with more severe symptoms mimicking ARs before initiation of treatment end up needing higher end-doses of MPH. It is important to notice that the ARs measured by the BSSERS-C may be 'influenced' by the global impairment level (CGI-S) of a patient. In this study, CGI-S proved to be a useful psychometric instrument, which confirmed its wide applicability in the psychiatry for measuring global impairment.

## Strength and limitations

The main strength of the study was the inclusion of a consecutive clinical and representative sample of children in their first medical treatment for ADHD in a 'real-life setting'. Moreover, the children were recruited from a well-defined catchment area, and the diagnosis of ADHD was based on best practice and routine examination using standardised instruments [29]. The included children represent a typical population of children with ADHD and one or several comorbid mental disorders, and with moderate to severe global impairment [89].

The clinical investigator performed all the weekly assessments of the same group of patients throughout the 12-week study period and monitored the IR-MPH treatment effects jointly with the clinician. Furthermore, the outcome measures of ADHD core symptoms were based on consensus ratings of all psychometric instruments in week 0 and 12. The intensive

monitoring of symptom reductions, ARs, and the careful titration of IR-MPH probably explains the low attrition rate.

The limitations of this observational study included lack of a control condition. Furthermore, the assessors could not be blinded to dose of MPH, because the dosing was individually titrated based on the weekly assessments of child's effects and ARs. Therefore, the symptom reductions of ADHD core symptoms and ARs due to IR-MPH treatment cannot be differentiated from the natural course of ADHD with a regression toward the mean and determination of the effect of the frequent contact with the clinical investigator. It can also be difficult to define a relevant cut-off for Nor or Bnor of ADHD core symptoms, and information might be lost when continuous variables (the ADHD sum scores) are dichotomised into cut-off values [90].

The study determined Nor and Bnor based on Inattention and Hyperactivity-Impulsivity of ADHD-RS-C using the existing Danish norms and cut-off values for ADHD-RS-P instead of the foreign norms and values for ADHD-RS-C [55, 63]. The WFIRS-P and TOVA are widely used but have not been validated in Denmark. The raw scores of TOVA were converted to $z$-scores based on week 0 total raw scores. BSSERS has no instructions for how to rate each item from 0 (*problem absent*) to 9 (*problem evokes serious impairment*) [49]. To overcome this, the authors of this study created a manual for ratings of the 17 items of BSSERS-C. Furthermore, the discriminative validity of BSSERS-C with regard to various mental and physical health problems and ARs are not known, and BSSERS-C has not been validated in Denmark.

The study was a short-term study which is a limitation. The extent to which the individually monitored long-term MPH treatment can improve the prognosis is sparsely elucidated. The efficacy of pharmacological treatment of ADHD in RCTs are often using fixed doses and short follow up time periods (<six month), whereas the long-term treatment effects are limited studied [15, 81]. But new long-term cohort studies have found pharmacological treatment associated to lower risk of accidents [91, 92], criminality [93], suicides [94], and better academical performance [22, 23, 95].

## Perspectives for the daily clinic

This study showed no causal relationship between baseline characteristics and symptom reductions or ARs of IR-MPH. Most patients continued treatment throughout the 12-weeks. The treatment outcome showed a favourable balance between symptom reductions and ARs. This is assuring compared to the recent debate of the efficacy of MPH [96, 97]. Our study also emphasizes that the clinicians should obtain baseline information about severity of ADHD symptoms and potential ARs, and they should continuously monitor these during up-titration of MPH using standardised rating scales. The BSSERS proved ineffective because it may not differentiate between symptoms and ARs, and there is an urgent need for development of a better instrument to monitor ARs of MPH in clinical practice. The parents seemed to be more aware of the severity of the patients' ADHD core symptoms than the teachers. Patients with severe global levels of symptoms and functional impairments appear to report more severe symptoms mimicking ARs before initiation of treatment and during the treatment period.

## Research perspectives

Future studies should investigate the importance of thorough individually titrated dosing of MPH, monitored on standard rating scales of ADHD core symptoms and ARs, and the effect of frequent brief contact with a clinician (e.g., a nurse) during the titration period. This will be important knowledge for the standard treatment of ADHD.

This study suggests an alignment of ways to measure and define a clinical response to MPH on a standard ADHD-rating scale and more valid methods to monitor ARs.

Studies of effects and adverse reactions to long-term treatment with methylphenidate are warrened as these observations are lacking in the literature.

## Supporting information

**S1 Fig. PRISMA flow chart of naturalistic observational clinical prospective studies of methylphenidate treatment of children with ADHD.**
(PDF)

**S1 Table. Criteria for literature search.**
(PDF)

**S2 Table. Included studies in the review of naturalistic observational clinical prospective studies of MPH treatment of drug naïve children with ADHD.**
(PDF)

**S3 Table. Comparison of ADHD core symptoms severity between included and excluded patients.**
(PDF)

**S4 Table. Baseline characteristics of included patients.**
(PDF)

**S5 Table. Distribution of number of patients with normalisation and borderline normalisation of clinician rated ADHD core symptoms score in week 0 and 12.**
(PDF)

**S6 Table. Absolute mean reduction score of ADHD core symptoms.**
(PDF)

**S7 Table. Further clinical follow-ups from week 0 to week 12.**
(PDF)

**S8 Table. Changes of adverse reactions.**
(PDF)

**S9 Table. Baseline characteristics and symptoms in week 0 of responder and nonresponder.**
(PDF)

**S1 File. Protocol INDICES.**
(PDF)

**S2 File. Protocol INDICES work package 6.**
(PDF)

## Acknowledgments

We wish to thank all children and their families for their participating in INDICES. Furthermore, we would like to thank the staff from Child and Adolescent Mental Health Centre: Anne Overgaard, Anne Torgny Andersen, Jørgen Dyrborg, Lone Kelkjær, Maj-Brit Åström, and Louise Hedegaard for help with including and examine the children, and Lars Clemmensen for the statistics as well as the INDICES Consortium.

## List of all partners in the INDICES Consortium

Lead author: Henrik Berg Rasmussen, henrik.berg.rasmussen@regionh.dk, Institute of Biological Psychiatry, Mental Health Centre Sct. Hans, The Capital Region of Denmark, Roskilde, Denmark.

Ditte Bjerre, Institute of Biological Psychiatry, Mental Health Centre Sct. Hans, The Capital Region of Denmark, Roskilde, Denmark. Majbritt Busk Madsen, Institute of Biological Psychiatry, Mental Health Centre Sct. Hans, The Capital Region of Denmark, Roskilde, Denmark. Laura Ferrero, Institute of Biological Psychiatry, Mental Health Centre Sct. Hans, The Capital Region of Denmark, Roskilde, Denmark. Kristian Linnet, Section of Forensic Chemistry, Department of Forensic Medicine, Faculty of Health Sciences, University of Copenhagen, Denmark. Ragnar Thomsen, Section of Forensic Chemistry, Department of Forensic Medicine, Faculty of Health Sciences, University of Copenhagen, Denmark. Gesche Jürgens, Unit of Clinical Pharmacology Roskilde University Hospital, Roskilde, Denmark. Kim Dalhoff, Department of Clinical Pharmacology, Bispebjerg University Hospital, Copenhagen, Denmark. Claus Stage, Department of Clinical Pharmacology, Bispebjerg University Hospital, Copenhagen, Denmark. Hreinn Stefansson, CNS Division, deCODE Genetics, Reykjavik, Iceland. Thomas Hankemeier, The Leiden/Amsterdam Center for Drug Research LACDR, Leiden University, Gorlaeus laboratories, Leiden, The Netherlands. Rima Kaddurah-Daouk, Department of Psychiatry and Behavioral Sciences, Duke University, Durham, NC, USA. Søren Brunak, Disease Systems Biology, Novo Nordisk Foundation Center for Protein Research, University of Copenhagen, Copenhagen, Denmark. Olivier Taboureau, Department of Systems Biology, Center for Biological Sequence Analysis, Technical University of Denmark, Lyngby, Denmark and INSERM, UMRS 973, MTi, Université Paris Diderot, Paris, France and Center for Biological Sequence Analysis, Technical University of Denmark, Kgs. Lyngby, Denmark. Grace Shema Nzabonimpa, Center for Biological Sequence Analysis, Technical University of Denmark, Kgs. Lyngby, Denmark. Tine Houmann, Child and Adolescent Mental Health Centre, Research Unit, Mental Health Services, The Capital Region of Denmark, Hellerup, Denmark. Pia Jeppesen, Child and Adolescent Mental Health Centre, Research Unit, Mental Health Services, The Capital Region of Denmark, Hellerup, Denmark. Kristine Kaalund-Brok, Child and Adolescent Mental Health Centre, Research Unit, Mental Health Services, The Capital Region of Denmark, Hellerup, Denmark. Peter Riis Hansen, Department of Cardiology, Copenhagen University Hospital, Hellerup, Denmark. Karl Emil Kristensen, Department of Cardiology, Copenhagen University Hospital, Hellerup, Denmark. Anne Katrine Pagsberg, Child and Adolescent Mental Health Centre, Research Unit, Mental Health Services, The Capital Region of Denmark, Hellerup, Denmark. Kerstin Plessen, Child and Adolescent Mental Health Centre, Research Unit, Mental Health Services, The Capital Region of Denmark, Hellerup, Denmark. Poul-Erik Hansen, Department of Science, Systems and Models, Roskilde University, Roskilde, Denmark. Wei Zhang, Department of Science, Systems and Models, Roskilde University, Roskilde, Denmark. Thomas Werge, Institute of Biological Psychiatry, Mental Health Centre Sct. Hans, Roskilde, Denmark. Jørgen Dyrborg, Child and Adolescent Mental Health Centre, The Capital Region of Denmark, Hillerød, Denmark. Maj-Britt Glenn Lauritsen, Child and Adolescent Mental Health Centre, The Capital Region of Denmark, Hillerød, Denmark.

## Author Contributions

**Conceptualization:** Tine Bodil Houmann, Henrik Berg Rasmussen, Pia Jeppesen.

**Data curation:** Kristine Kaalund-Brok, Tine Bodil Houmann.

**Formal analysis:** Kristine Kaalund-Brok, Morten Aagaard Petersen, Jens Richardt Møllegaard Jepsen, Pia Jeppesen.

**Funding acquisition:** Henrik Berg Rasmussen.

**Investigation:** Kristine Kaalund-Brok, Tine Bodil Houmann, Marie Bang Hebsgaard, Maj-Britt Glenn Lauritsen, Louise Hyldborg Lundstrøm, Helene Grønning, Lise Darling, Susanna Reinert-Petersen.

**Methodology:** Kristine Kaalund-Brok, Tine Bodil Houmann, Henrik Berg Rasmussen, Pia Jeppesen.

**Project administration:** Kristine Kaalund-Brok, Tine Bodil Houmann, Pia Jeppesen.

**Resources:** Tine Bodil Houmann, Marie Bang Hebsgaard, Maj-Britt Glenn Lauritsen, Louise Hyldborg Lundstrøm, Helene Grønning, Susanna Reinert-Petersen.

**Software:** Morten Aagaard Petersen.

**Supervision:** Anne Katrine Pagsberg, Kerstin Jessica Plessen, Henrik Berg Rasmussen, Pia Jeppesen.

**Validation:** Kristine Kaalund-Brok, Morten Aagaard Petersen, Jens Richardt Møllegaard Jepsen.

**Writing – original draft:** Kristine Kaalund-Brok.

**Writing – review & editing:** Tine Bodil Houmann, Marie Bang Hebsgaard, Maj-Britt Glenn Lauritsen, Louise Hyldborg Lundstrøm, Helene Grønning, Lise Darling, Susanna Reinert-Petersen, Morten Aagaard Petersen, Jens Richardt Møllegaard Jepsen, Anne Katrine Pagsberg, Kerstin Jessica Plessen, Henrik Berg Rasmussen, Pia Jeppesen.

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
