## [Decision Letter · Decision Letter 0]

27 Oct 2020

PONE-D-20-01564

Outcomes of a 12-week ecologically valid observational study of first treatment with methylphenidate in a representative clinical sample of drug naïve children with ADHD.

PLOS ONE

Dear Dr. Kaalund-Brok,

Thank you for submitting your manuscript to PLOS ONE. After careful consideration, we feel that it has merit but does not fully meet PLOS ONE’s publication criteria as it currently stands. Therefore, we invite you to submit a revised version of the manuscript that addresses the points raised during the review process.

Please address carefully all the extensive list of amendments and clarifications requested by the reviewers in the various sections of the manuscript. 

We look forward to receiving your revised manuscript.

Kind regards,

Andrea Martinuzzi

Academic Editor

PLOS ONE

Journal Requirements:

3. One of the noted authors is a group or consortium [INDICES]. In addition to naming the author group and listing the individual authors and affiliations within this group in the acknowledgments section of your manuscript, please also indicate clearly a lead author for this group along with a contact email address.

5. Please include a copy of Table 10 which you refer to in your text on page 20.

Reviewers' comments:

Reviewer's Responses to Questions

**Comments to the Author**

1. Is the manuscript technically sound, and do the data support the conclusions?

Reviewer #1: Partly

Reviewer #2: Yes

Reviewer #3: Partly

2. Has the statistical analysis been performed appropriately and rigorously? 

Reviewer #1: No

Reviewer #2: Yes

Reviewer #3: Yes

3. Have the authors made all data underlying the findings in their manuscript fully available?

Reviewer #1: Yes

Reviewer #2: Yes

Reviewer #3: Yes

4. Is the manuscript presented in an intelligible fashion and written in standard English?

Reviewer #1: Yes

Reviewer #2: Yes

Reviewer #3: Yes

5. Review Comments to the Author

Reviewer #1: The manuscript entitled ‘Outcomes of a 12-week ecologically valid observational study of first treatment with methylphenidate in a representative clinical sample of drug naïve children with ADHD’ with the aim to describe treatment responses and their predictors during the first 12 weeks of MPH treatment using repeated measurements of symptoms and adverse reactions (ARs) to treatment in 207 children recently diagnosed with ADHD.

The manuscript can be further improved based on the following comments.

Study Design

Page 6 the sentence ‘There was made a protocol (S11 and S12)’ to be revised.

Page 7, the statement ‘the study was not registered in a registry for clinical trials (see the ethics section)’ not clear. In the ethics section, it was written ‘The study was registered in ClinicalTrials.gov (NCT04366609)’

CGI-S, CGI-I, WFIRS-P, BSSERS-C language version and scoring method including classification of scores to be highlighted.

Page 11, for the statement ‘Ten percent missing data on items on WFIRS-P subscales were allowed and missing data on items were set as 0 (never or not at all)’ reason to be provided.

Description on the missing data to be provided. i.e percentages of missing data, pattern etc.

Page 11 symbol X² to be replaced with symbol 'Chi' square. Likewise with S9 Table footnote (italicize X to be replaced with 'Chi' symbol).

Page 12, the citation for SPSS including publisher name to be highlighted.

Results

Page 12 for ‘187 (90.3%) patients (mean age 9.6 (SD 1.5)’ the mean and SD are similar to mean, sd for 207 patients.

Figure 1, difficult to be visualized.

Page 13 Table 1, decimal points to be standardized. 9.88 (comma to be replaced with dot)

Page 16 Table 2, bp/m to be written as bpm. Effect size could be presented.

Page 15 for the statement ‘The mean percent reductions of scores on the clinician-rated ADHD-RS-C subscale from week 0 to 12 were 52.0% on Inattention and 56.0% on Hyperactivity-Impulsivity' 52.0 % and 56.0% to be revised as 51.7% and 53.0%.

Page 15, for the statement ‘ nd the mean percentage reductions on the parent-rated ADHD-RS-P subscales were 48.1% on Inattention, 45.0% on Hyperactivity-Impulsivity, and 50.7% on Conduct problems (S6 Table)’ the figures did not tally with the figures in S6Table.

Page 15, for ‘CI 95% 2.2 to 2.5’ & Page 17 ‘CI 95% 0.5 to 0.9), Page 19 & 21 text and Table 4 & 5' CI95% to be written as 95%CI.

Page 17, Adverse reactions, Table 2 to be cited for height and weight findings.

Page 18 Table 3, what estimate refers, to be clearly labelled/defined.

Page 18 Table 3, Page 20 Table 4, Page 21 Table 5, model fit/goodness of fit measures to be stated.

Page 21 Table 5, the estimate to be replaced with b. SD errors to be written as std error.

Page 21, for the statement 'Severely ill and most extremely ill rated patients (CGI-S 6-7) had a significantly higher risk of reduced appetite compared with patients rated not ill to moderately ill (CGI-S 1-5) in week 0 and throughout the whole treatment period (OR = 2.41, CI 95% 1.42 to 4.08).' the results to be displayed in the table.

Page 21, for the statement 'Higher CGI-S scores at week 0 predicted a higher end dose of IR-MPH. CGI-S at week 0 and age explained together 12.7% (p > 0.001) of the variance of the end-dose IR-MPH' indicate clearly R square and denoted in the table footnote. For p >0.001, to use < and representative p value.

S4 Table footnote, were all these confirmed missing data and not numbers after minus the missing data? Missing data: ¹ n = 147, ² n= 143, ³ n = 195, ⁴ n = 193; S5 Table footnote Missing data: ¹ n = 66, ² n = 48, ³ n = 59; S6 Table footnote Missing data: ¹ n = 174, ² n = 171, ³ n = 163, ⁴ n = 179, ⁵ n = 127, ⁶. Likewise please check S7 Table and S8 Table.

S4 Table, for 'Medium, higher education (15-14 years)' should be written as 15-16 years.

S4 Table, Cap for 'T'otal. Likewise for total in S7 Table.

S9 Table 9, symbol % to be highlighted in the table.

S5 Table, symbol % for individual figure to be omitted since the symbol was highlighted after the variable name. Likewise with S6 Table.

Limitation on sample size or power of study to be discussed if any.

Reviewer #2: This is a timely and well conducted study. The rationale is presented in a very clear way and the authors make good points in the need for more echologic valid studies as well as the lack of a standard definition of "response" in the field. the methods are appropriate and the reporting good.. I only have a few suggestions:

“Attention deficit/hyperactivity disorder (ADHD) is a heterogeneous neurodevelopmental disorder characterised by pervasive and impairing symptoms of inattention and/or hyperactivity and impulsivity with onset of symptoms before age 7 years (International Classification of Diseases and Related Health Problems, ICD-10)[1]” is per se not correct as ICD -10 does not include ADHGD; rather, it includes Hyperkinetic syndrome, which is roughly equivalent to the ADHD combined presentation of the DSM-5. So the authors may rephrase as follows (or similar): “Attention deficit/hyperactivity disorder (ADHD) as defined in the DSMS is a heterogeneous neurodevelopmental disorder characterised by pervasive and impairing symptoms of inattention and/or hyperactivity and impulsivity with onset of symptoms before age 12 or 7 when considering the definition of Hyperkinetic syndrome (International Classification of Diseases and Related Health Problems, ICD-10)[1], roughly equivalent to the combined presentation of ADHD as per the DSM”

The focus of the paper ins on the pharmacological treatment … but the author may want to remind the reader that non pharmacological treatments, in particular parent traingin behaviors, offer complementary interventions (eg, for opposition behaviors) (eg: https://pubmed.ncbi.nlm.nih.gov/29083042/ )

“Several meta-analyses of the short-term efficacy of immediate release MPH (IR-MPH) in randomized placebo-controlled trials have reported large effect sizes (range from 0.54 to 1.83) on ADHD core symptoms, when effects are measured as differences in endpoint or change in scores of parent-rated and teacher-rated ADHD symptoms and behaviour”: please provide references

Methods

“There was made a protocol (S11 and S12)” is a bit odd; I would suggest. Study protocol is available (S11 and S12)

Results

I wonder whether the normalisation, in addition to /borderline normalisation results could be also presented (including in the abstract)

Discussion

The lack of blinding should also be discussed among the limitations

Reviewer #3: The paper focuses on one of the most common psychiatric diseases in childhood and adolescence. This fact makes it particularly interesting and current. The drug therapy of ADHD and the risk / benefit relationship between therapy and adverse reactions are a permanent topic of discussion in the scientific community. Therefore this paper addresses the two most controversial points: the therapeutic efficacy of methylphenidate and its safety.

Clinicians, statisticians and epidemiologists consider time to be one of the main variables in observational and experimental studies. From this point of view, the choice to make an ecological study of only 12 weeks creates some perplexity. One of the critical points of ADHD drug therapy is maintaining efficacy over time. In the MTA study, also cited by the authors of this paper, a decrease in the therapeutic response was observed between the second and third year of treatment. The safety evaluation of a drug also takes a long time, especially for the observation of rare adverse events.

The study design and methodology used are very good and the statistical analysis is of a remarkable level. The choice of predictors of the therapeutic response is accurate. The sample size is adequate for the objectives of the study.

6. PLOS authors have the option to publish the peer review history of their article (what does this mean?). If published, this will include your full peer review and any attached files.

Reviewer #1: No

Reviewer #2: No

Reviewer #3: **Yes: **Pietro Panei

---

## [Author Response · Author response to Decision Letter 0]

24 Feb 2021

Dear Andrea Martinuzzi, Academic Editor, PLOS ONE

Thank you for the opportunity to resubmit our manuscript for further consideration in PLOS ONE. We are very thankful for the peer reviewer’s effort and time taken to review our manuscript. We very much appreciate the comments and the suggestions that have helped us clarify certain points and further improve the quality of our manuscript. Please find below our response to the reviewer comments.

We hope that the revised version is now acceptable for publication in PLOS One and look forward to hearing from you.

Sincerely,

Kristine Kaalund-Brok, 

on behalf of the group of authors 

We begin by addressing the following prompts:

1. The manuscript meets PLOS ONE's style requirements. 

2. We include a Data Availability statement.

3. One of the noted authors is a group or consortium [INDICES]. A lead author for this group along with a contact email address is now provided on the top of “List of all partners in the INDICES Consortium”.

4. All data are available in the text, and we long longer use the term “data not shown”.

5. The reference to Table 10 was a mistake, it is now corrected to Table 4.

---

## [Decision Letter · Decision Letter 1]

15 Apr 2021

PONE-D-20-01564R1

Outcomes of a 12-week ecologically valid observational study of first treatment with methylphenidate in a representative clinical sample of drug naïve children with ADHD.

PLOS ONE

Dear Dr. Kaalund-Brok,

Thank you for submitting your manuscript to PLOS ONE. After careful consideration, we feel that it has merit but does not fully meet PLOS ONE’s publication criteria as it currently stands. Therefore, we invite you to submit a revised version of the manuscript that addresses the points raised during the review process.

Please address the few remaining concerns of reviewer 2.

We look forward to receiving your revised manuscript.

Kind regards,

Andrea Martinuzzi

Academic Editor

PLOS ONE

Journal Requirements:

Reviewers' comments:

Reviewer's Responses to Questions

**Comments to the Author**

1. If the authors have adequately addressed your comments raised in a previous round of review and you feel that this manuscript is now acceptable for publication, you may indicate that here to bypass the “Comments to the Author” section, enter your conflict of interest statement in the “Confidential to Editor” section, and submit your "Accept" recommendation.

Reviewer #1: (No Response)

Reviewer #2: (No Response)

Reviewer #3: All comments have been addressed

2. Is the manuscript technically sound, and do the data support the conclusions?

Reviewer #1: Partly

Reviewer #2: Yes

Reviewer #3: Yes

3. Has the statistical analysis been performed appropriately and rigorously? 

Reviewer #1: Yes

Reviewer #2: Yes

Reviewer #3: Yes

4. Have the authors made all data underlying the findings in their manuscript fully available?

Reviewer #1: No

Reviewer #2: Yes

Reviewer #3: Yes

5. Is the manuscript presented in an intelligible fashion and written in standard English?

Reviewer #1: Yes

Reviewer #2: Yes

Reviewer #3: Yes

6. Review Comments to the Author

Reviewer #1: (No Response)

Reviewer #2: I would suggest only a couple of points to improve the quality of the paper:

" Several meta-analyses of the short-term efficacy of immediate release MPH (IR-MPH) in randomized placebo-controlled trials

have reported large effect sizes (range from 0.54 to 1.78) on ADHD core symptoms, when effects are measured as differences

in endpoint or change in scores of parent-rated and teacher-rated ADHD symptoms and behaviours[14,15].: Network meta-analyses (eg, https://pubmed.ncbi.nlm.nih.gov/30097390/ and https://pubmed.ncbi.nlm.nih.gov/28700715/ ) should also be mentioned as their estimates are more precise than those form pairwise meta-analyses

in the discussion, the authors may refer to effects of MPH on important outcomes not generally include din RCTs, such as physical injuries (https://pubmed.ncbi.nlm.nih.gov/31302218/), suicides, etc...a summary is reported in https://pubmed.ncbi.nlm.nih.gov/32905677/

Reviewer #3: Dear authors,

Congratulations on the great deal of work you have done. Your paper is original and interesting. Furthermore, the fact that there is no control group does not represent a limitation since an observational field study. Observational studies designed well and conducted according to the protocol can give the same information as experimental clinical trials. If anything, a short follow-up is unusual (12 weeks) for this type of study. Having available the administrative and clinical data of the observed cohort would be interesting to observe it for a longer period. The method used to define the predictors of the methods of treatment with methylphenidate is elegant, appropriate and very sophisticated. It is not clear the reason for choosing numerous primary objectives also since it is a multifactorial syndrome. It would be better to identify one or two main outcomes of pharmacological treatment and analyzing data in relation to these primary outcomes. In particular it is important to know the relationship between cost and benefit of treatment with methylphenidate even if there are numerous studies on this matter. Furthermore, predicters analysis is easily applicable to clinical practice? Does it allow you to change the natural history of the syndrome? In the future it would be useful to conduct a deepening also in this field. The figures and tables are clear and help in reading the paper, especially the flowchart allows you to immediately understand the design of the study.

Overall you have done a good work of a high level of statistical and a study of the factors involved in the ADHD useful for understanding the multifatorial mechanisms that are at the origin of the syndrome.

This paper does not violate the ethics code of scientific publications.

7. PLOS authors have the option to publish the peer review history of their article (what does this mean?). If published, this will include your full peer review and any attached files.

Reviewer #1: No

Reviewer #2: No

Reviewer #3: No

---

## [Author Response · Author response to Decision Letter 1]

31 May 2021

Comments to the Author

1. If the authors have adequately addressed your comments raised in a previous round of review and you feel that this manuscript is now acceptable for publication, you may indicate that here to bypass the “Comments to the Author” section, enter your conflict of interest statement in the “Confidential to Editor” section, and submit your "Accept" recommendation.

Reviewer #1: (No Response)

Reviewer #2: (No Response)

Reviewer #3: All comments have been addressed

2. Is the manuscript technically sound, and do the data support the conclusions?

Reviewer #1: Partly

Reviewer #2: Yes

Reviewer #3: Yes

3. Has the statistical analysis been performed appropriately and rigorously? 

Reviewer #1: Yes

Reviewer #2: Yes

Reviewer #3: Yes

4. Have the authors made all data underlying the findings in their manuscript fully available?

Reviewer #1: No

Reply: The study measured the outcomes of a 12-week ecologically valid observational study of first treatment with methylphenidate in a representative clinical sample of drug naïve children with ADHD. The study was a part of the routine care in the clinic. The data were both documented in the medical record at the Child and Adolescent Mental Health Centre and in an anonymized data file with a separate key file. All data were sensitive patient information. The pseudonymous individual participant data that underlie the results reported in this article, (text, tables, figures, and appendices) can be made available to investigators for individual participant data meta-analyses that have been approved by independent review committees.

Reviewer #2: Yes

Reviewer #3: Yes

5. Is the manuscript presented in an intelligible fashion and written in standard English?

Reviewer #1: Yes

Reviewer #2: Yes

Reviewer #3: Yes

6. Review Comments to the Author

Reviewer #1: (No Response)

Reviewer #2: I would suggest only a couple of points to improve the quality of the paper:

" Several meta-analyses of the short-term efficacy of immediate release MPH (IR-MPH) in randomized placebo-controlled trials

have reported large effect sizes (range from 0.54 to 1.78) on ADHD core symptoms, when effects are measured as differences

in endpoint or change in scores of parent-rated and teacher-rated ADHD symptoms and behaviours[14,15].: Network meta-analyses (eg, https://pubmed.ncbi.nlm.nih.gov/30097390/ and https://pubmed.ncbi.nlm.nih.gov/28700715/ ) should also be mentioned as their estimates are more precise than those form pairwise meta-analyses

Reply: Thank you very much, the references are now added to the introduction.

Also, two network metanalyses have reported favorable efficacy of MPH compared with placebo in randomized placebo-controlled trials[18][19].

in the discussion, the authors may refer to effects of MPH on important outcomes not generally include din RCTs, such as physical injuries (https://pubmed.ncbi.nlm.nih.gov/31302218/), suicides, etc...a summary is reported in https://pubmed.ncbi.nlm.nih.gov/32905677/ . 

Reply: Again, thank you very much. The theme and references are now added to the strength and limitation section.

The study was a short-term study whis is a limitation. The extent to which the individually monitored long-term MPH treatment can improve the prognosis is sparsely elucidated. The efficacy of pharmacological treatment of ADHD in RCTs are often using fixed doses and short follow up time periods (<six month), whereas the long-term treatment effects are limited studied[15,81]. But new long-term cohort studies have found pharmacological treatment associated to lower risk of accidents[91,92], criminality[93], suicides[94], and better academical performance[22,23,95].

Reviewer #3: Dear authors,

Congratulations on the great deal of work you have done. Your paper is original and interesting. Furthermore, the fact that there is no control group does not represent a limitation since an observational field study. Observational studies designed well and conducted according to the protocol can give the same information as experimental clinical trials. If anything, a short follow-up is unusual (12 weeks) for this type of study. Having available the administrative and clinical data of the observed cohort would be interesting to observe it for a longer period. The method used to define the predictors of the methods of treatment with methylphenidate is elegant, appropriate and very sophisticated. It is not clear the reason for choosing numerous primary objectives also since it is a multifactorial syndrome. It would be better to identify one or two main outcomes of pharmacological treatment and analyzing data in relation to these primary outcomes. In particular it is important to know the relationship between cost and benefit of treatment with methylphenidate even if there are numerous studies on this matter. Furthermore, predicters analysis is easily applicable to clinical practice? Does it allow you to change the natural history of the syndrome? In the future it would be useful to conduct a deepening also in this field. The figures and tables are clear and help in reading the paper, especially the flowchart allows you to immediately understand the design of the study.

Overall you have done a good work of a high level of statistical and a study of the factors involved in the ADHD useful for understanding the multifatorial mechanisms that are at the origin of the syndrome.

This paper does not violate the ethics code of scientific publications.

Reply: 

Thank you very much for the advises to review this article and for the compliments to the study. 

It is correct that a follow-up time of 12-weeks is short-term follow-up especially in an observational study. As mentioned earlier, the current paper present data regarding the 12 weeks follow-up, while the next report from this study will focus on long-term outcome, including register-based prescription data to describe the continuation/discontinuation of MPH use over a 3-year follow-up period. 

The purpose of this study was to detect the MPH treatment outcome (including predictors) in a broad sense with regard to the multifactorial origin, the diagnostic criteria of ADHD, and the guidelines for MPH treatment of ADHD. The aim was not to look at cost and benefit of MPH treatment, but ofcause it is an interesting research question.

7. PLOS authors have the option to publish the peer review history of their article (what does this mean?). If published, this will include your full peer review and any attached files.

Do you want your identity to be public for this peer review? For information about this choice, including consent withdrawal, please see our Privacy Policy.

Reviewer #1: No

Reviewer #2: No

Reviewer #3: No

---

## [Editor Report · Decision Letter 2]

14 Jun 2021

Outcomes of a 12-week ecologically valid observational study of first treatment with methylphenidate in a representative clinical sample of drug naïve children with ADHD.

PONE-D-20-01564R2

Dear Dr. Kaalund-Brok,

We’re pleased to inform you that your manuscript has been judged scientifically suitable for publication and will be formally accepted for publication once it meets all outstanding technical requirements.

Kind regards,

Andrea Martinuzzi

Academic Editor

PLOS ONE
---

## [Editor Report · Acceptance letter]

9 Jul 2021

PONE-D-20-01564R2 

Outcomes of a 12-week ecologically valid observational study of first treatment with methylphenidate in a representative clinical sample of drug naïve children with ADHD. 

Dear Dr. Kaalund-Brok:

I'm pleased to inform you that your manuscript has been deemed suitable for publication in PLOS ONE. Congratulations! Your manuscript is now with our production department. 

Kind regards, 

on behalf of

Dr. Andrea Martinuzzi 

Academic Editor

PLOS ONE